# A sulfide-sensor and a sulfane sulfur-sensor collectively regulate sulfur-oxidation for feather degradation by *Bacillus licheniformis*

Chao Tang[1,4], Jingjing Li[1,3,4], Yuemeng Shen[1], Menghui Liu[1], Honglei Liu[1], Huaiwei Liu ⬤ [1], Luying Xun ⬤ [1,2✉] & Yongzhen Xia ⬤ [1✉]

*Bacillus licheniformis* MW3 degrades bird feathers. Feather keratin is rich in cysteine, which is metabolized to produce hazardous sulfide and sulfane sulfur. A challenge to *B. licheniformis* MW3 growing on feathers is to detoxify them. Here we identified a gene cluster in *B. licheniformis* MW3 to deal with these toxicity. The cluster contains 11 genes: the first gene *yrkD* encodes a repressor, the 8th and 9th genes *nreB* and *nreC* encode a two-component regulatory system, and the 10th and 11th genes encode sulfide: quinone reductase (SQR) and persulfide oxygenase (PDO). SQR and PDO collectively oxidize sulfide and sulfane sulfur to sulfite. YrkD sensed sulfane sulfur to derepress the 11 genes. The NreBC system sensed sulfide and further amplified the transcription of *sqr* and *pdo*. The two regulatory systems synergistically controlled the expression of the gene cluster, which was required for the bacterium to grow on feather. The findings highlight the necessity of removing sulfide and sulfane sulfur during feather degradation and may help with bioremediation of feather waste and sulfide pollution.

[1] State Key Laboratory of Microbial Technology, Shandong University, Qingdao 266237, People's Republic of China. [2] School of Molecular Biosciences, Washington State University, Pullman, WA 99164-7520, USA. [3] Present address: Institut für Mikrobiologie & Biotechnologie of Rheinische Friedrich-Wilhelms-Universität Bonn, Bonn, Germany. [4] These authors contributed equally: Chao Tang, Jingjing Li. ✉email: luying_xun@vetmed.edu.edu; xiayongzhen2002@sdu.edu.cn

The gram-positive bacterium *Bacillus licheniformis* MW3 is common on bird feathers[1] and has been used to degrade feather waste into protein feeds for animals[2,3]. Feather keratin is rich in cysteine residues[4], which were linked by disulfide bonds with the formation of tightly packed structures to resist chemical and physical factors[5,6]. When *B. licheniformis* MW3 degrades feathers, it may need to disrupt the disulfide bonds, and deal with the stress from released L-cysteine and sulfide ($H_2S$ and $HS^-$). L-cysteine even at low concentrations is cytotoxic to prokaryotes and eukaryotes[7,8], and its metabolism releases sulfide[9–11]. Although $H_2S$ is a common metabolic product of heterotrophic bacteria[10], it is toxic at elevated levels, as it inhibits cytochrome c oxidation in the aerobic electron transport chain[12,13]. It is unclear how *B. licheniformis* MW3 copes with the self-produced $H_2S$ during feather degradation.

Besides being oxidized by chemolithotrophic bacteria, sulfide is also oxidized by numerous heterotrophic bacteria and mitochondria of certain eukaryotes, including humans, by using two key enzymes: sulfide: quinone oxidoreductase (SQR) and persulfide dioxygenase (PDO)[14,15]. SQR oxidizes self-produced and exogenous sulfide to zero-valent sulfur ($S^0$) which is further oxidized by PDO to sulfite[10,16]. $S^0$ comes in different forms in the cytoplasm, including persulfide (HSSH and RSSH), polysulfide ($H_2S_n$ and $RS_nH$, n > 2), and elemental sulfur persulfide, which are collectively referred as sulfane sulfur[17]. Sulfite spontaneously reacts with sulfane sulfur to produce thiosulfate, and rhodanese (RHOD) may speed up the reaction[15,18,19]. The genes encoding SQR and PDO are present in *B. licheniformis* MW3[10], and they may help the bacterium to degrade feathers.

$H_2S$ is also a gaseous signaling molecule, that regulates numerous biological activities in mammals[9,20–22]. Although it has been suggested as a gasotransmitter in bacteria, available evidence suggests that $H_2S$ is first oxidized to sulfane sulfur, which is sensed by gene regulators in bacteria[23]. Several gene regulators that repress the expression of sulfur-oxidizing genes in bacteria sense sulfane sulfur but not sulfide[24–26]. CstR and CsoR are homologous regulators that repress the genes involved in sulfur oxidation in *Staphylococcus aureus* and *Streptomyces coelicolor*, respectively[27,28]. When they are modified by sulfane sulfur, the repressed genes are derepressed. SqrR specifically senses sulfane sulfur and derepresses the genes related to sulfide-dependent photosynthesis in the purple photosynthetic bacterium *Rhodobacter capsulatus*[24,29]. FisR is an activator that senses sulfane sulfur and activates the expression of the genes for sulfur oxidation in *Cupriavidus pinatubonensis* JMP134[30]. Further, several gene regulators including OxyR, MexR, MgrA, and MarR that are known to respond to $H_2O_2$ also respond to sulfane sulfur[27,31–33]. The cysteine thiols are modified by sulfane sulfur in these proteins.

Here, we report that *B. licheniformis* MW3 used SQR and PDO to detoxify $H_2S$ and sulfane sulfur, which was required for the bacterium to grow on feathers. Their genes were regulated by two regulatory systems: a CstR-like repressor YrkD that sensed sulfane sulfur, and a two-component system (TCS) NreBC that sensed sulfide. These genes with several other genes formed a gene cluster, which was transcribed from at least two overlapping promoters. An operon of *sqr* and *pdo* that is regulated by two gene regulators have not been reported to date. A fine-tuned regulation of sulfide oxidation by the two regulating systems helped cells to deal with the toxic effects of sulfane sulfur and $H_2S$. Further, NreB was shown as a $H_2S$ sensor, whereas previously reported regulators controlling sulfide-oxidizing genes only respond to sulfane sulfur.

## Results

### A gene cluster of 11 genes in *B. licheniformis* MW3 was involved in sulfur metabolism. 
Besides *sqr* and *pdo* in *B. licheniformis* MW3, nine upstream genes are also involved in sulfur metabolism (Fig. 1a). These 11 genes formed a gene cluster. The cluster started with *yrkD* that encodes a regulatory protein with 50.0% sequence identity to $CstR_{Sa}$, a repressor of sulfur-oxidizing genes in *Staphylococcus aureus*[26,28]. The other eight genes were *yrkE*, *yrkF*, *yrkH*, *yrkI*, *yrkJ*, *ydfQ*, *nreB*, and *nreC* (Fig. 1a). YrkE and YrkI are homologs of *dsrE* and *tusA*, respectively. They are potential sulfane sulfur carriers[34,35]; YrkF is a RHOD-like protein that could receive sulfane sulfur from YrkE and YrkI, and transfer to potential receptors[34]; YrkH is another hypothetical PDO; YrkJ is a hypothetical TauE that is involved in sulfite transport[36]; YdfQ is hypothetical thioredoxin. NreBC forms a TCS, consisting of a sensor histidine kinase and a response regulator, and their genes were followed by *sqr* and *pdo*. Of the 11 genes, YrkD and NreBC were involved in gene regulation.

We noticed a TCS that potentially regulates sulfide oxidation. Curiously, other potential TCSs were screened from three genes upstream and downstream of the 454 *sqr-pdo* operons, which were reported previously[10]. Twelve pairs of genes encoding TCSs were identified from Gram-positive bacteria (Supplementary Data 1). They were present in the Genera of *Bacillus* (six species), *Geobacillus* (four species), *Mycobacterium* (one species), and *Alicyclobacillus* (one species). The NreB and NreC proteins of *B. licheniformis* MW3 had sequence identities at >65% to other NreB and NreC proteins in the genus *Bacillus* (Fig. 1b) and at ~50% to related NreB and NreC proteins in the genus *Geobacillus* (Fig. 1b). Phylogeny analysis showed that these sensor histidine kinases and response regulators were coevolved (Supplementary Fig. 1a, b), and their phylogenetic structures were similar to those from the analysis of their 16 S rDNA sequences (Fig. 1b).

Most histidine kinases contained a PAS domain except the one from *Mycobacterium vanbaalenii* PYR-1. These PAS domains were conserved and formed a separate clade from the known PAS domains that harbor either a Fe-S cluster or a heme B via phylogenetic analysis (Fig. 1c). The results suggest that these histidine kinases could sense unknown signals.

### The functions of the gene cluster in *B. licheniformis* MW3 during feather degradation. 
*B. licheniformis* MW3 and its mutants Δ*yrkD*, Δ*nreBC*, Δ*sqr-pdo*, and Δ*yrkE-ydfQ* were grown in the feather medium. MW3 grew well, Δ*yrkD* grew slightly better than MW3, other mutants grew poorly on feather (Fig. 2a). Among the three poorly growing mutants, Δ*yrkE-ydfQ* grew better than Δ*nreBC* and Δ*sqr-pdo* (Fig. 2a). During feather degradation by MW3 for 4 days, cysteine and thiosulfate were gradually increased to 5.4 mM and 274 μM (Fig. 2b), respectively. Both sulfide and sulfite was not detected. We tested cysteine stress on the growth of *B. licheniformis* MW3 and its mutants by cysteine in an LB medium. Cysteine from 0.4–5 mM displayed increased inhibition to MW3 (Supplementary Fig. 2a). At 0.6 mM, cysteine was inhibitory to MW3, and the mutants Δ*nreBC*, Δ*sqr-pdo*, and Δ*yrkE-ydfQ* grew more poorly than MW3 did, but the Δ*yrkD* mutant grew better than MW3 did (Supplementary Fig. 2b). On LB plates with 3 μL of 8 M cysteine spotted in the middle, the Δ*yrkD*, Δ*nreBC*, Δ*sqr-pdo*, and Δ*yrkE-ydfQ* mutants were more sensitive to cysteine, generating an inhibition zone that was more than 1.7, 3.6, 3.7, and 3.7 times larger than that of MW3 (Supplementary Fig. 2c, d). The results indicate that this gene cluster helps the bacterium to grow on feathers and to resist cysteine stress.

The mechanism of cysteine toxicity was tested. Resting cells of *B. licheniformis* MW3 at an $OD_{600}$ of 2.0 were incubated with 600 μM cysteine. The cells produced 318 μM, 589 μM, and 376 μM sulfane sulfur at 5 min, 60 min, and 90 min (Fig. 2c), and they produced 10 μM, 56 μM, and 74 μM sulfide at 5 min, 60 min, and 90 min (Fig. 2d). Both sulfide and sulfane sulfur are

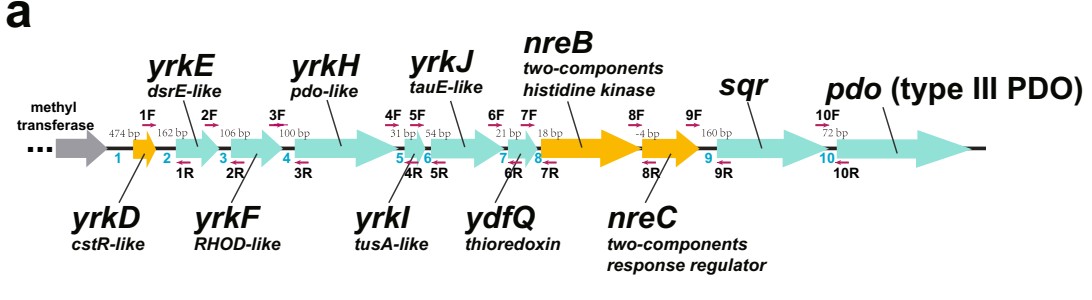

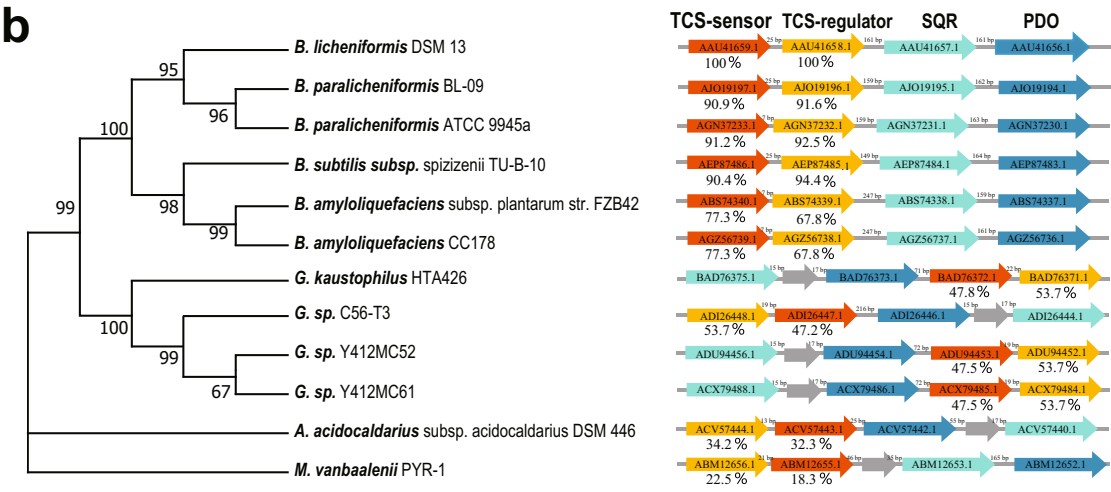

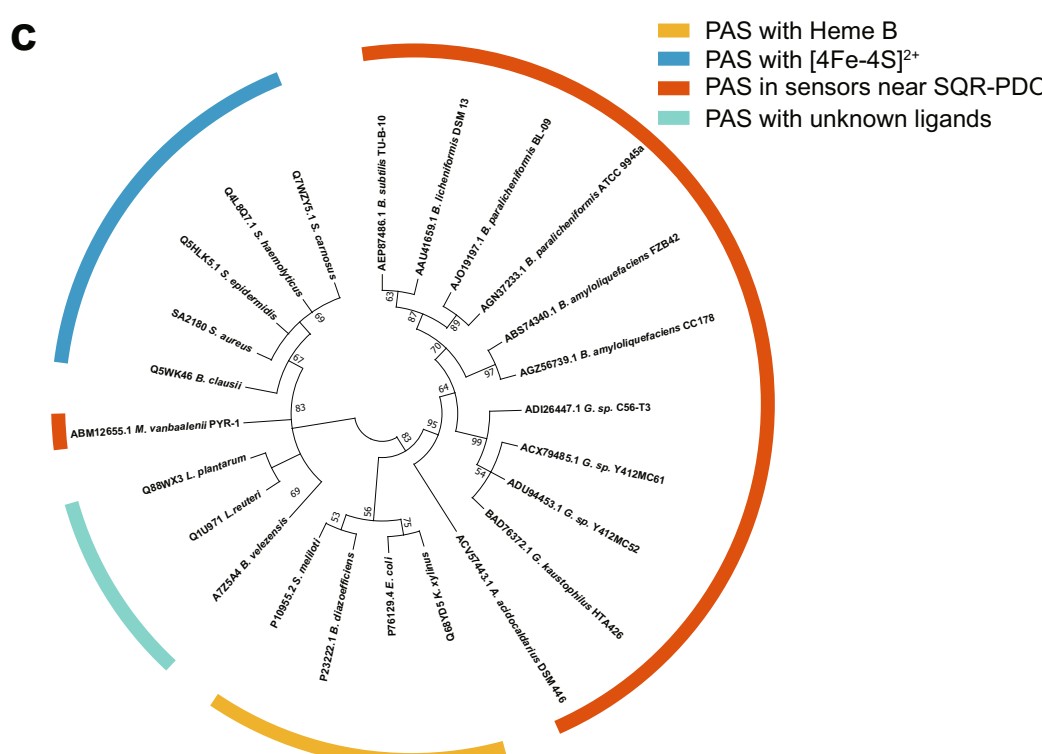

potentially toxic[37]. The toxicity of NaHS and elemental sulfur to *B. licheniformis* MW3 was tested in LB medium with 50, 100, and 200 μM NaHS or with 20, 40, and 60 μM elemental sulfur. 50 μM NaHS and 40 μM elemental sulfur were slightly inhibitory to MW3, but 100 μM NaHS and 60 μM elemental sulfur strongly

slowed down the growth (Supplementary Fig. 3a, b). The four tested mutants (Δ*yrkD*, Δ*nreBC*, Δ*sqr-pdo*, and Δ*yrkE-ydfQ*) all grew more poorly than MW3 did in the presence of 50 μM sulfide (Fig. 2e). 40 μM sulfane sulfur inhibited the cell growth of all mutants except Δ*nreBC* (Fig. 2f). Thus, *B. licheniformis* MW3

**Fig. 1 The annotation of a gene containing *sqr* and *pdo* in *B. licheniformis*. a** The gene cluster encoding $H_2S$ oxidation in *B. licheniformis* MW3. The arrow on a black line indicated the genes in the chromosome. The gene names, related homologs, and gene IDs were given. The red arrow was used to indicate the primers designed for co-transcription analysis (Fig. 4a and Supplementary Fig. 4). The intergenic regions among the gene cluster were numbered with Arabic numbers in blue color from IR1 to IR10. **b** The phylogenetic tree of 16 S rDNA of 12 bacteria that have TCS genes within 3 loci of an *sqr-pdo* operon. The locations of the TCS genes, *sqr*, and *pdo* in the chromosome of the corresponding strains. The arrow indicated the genes, and the related protein IDs were given in the arrows. The percentages of protein identities of NreB and NreC of *B. licheniformis* DSM 13 with the corresponding proteins in other bacteria were shown under the genes. Sequence alignment was done by using ClustalW, and the trees were built by using MEGA X with the Maximum Likelihood method and Tamura-Nei model. **c** Phylogeny analysis of the PAS domains encoded in the sensor genes with known PAS domains. The branches were colored, and their functions were noted. Sequence alignment was done by using ClustalW, and the tree was built by using MEGA X with the Neighbor-Joining method.

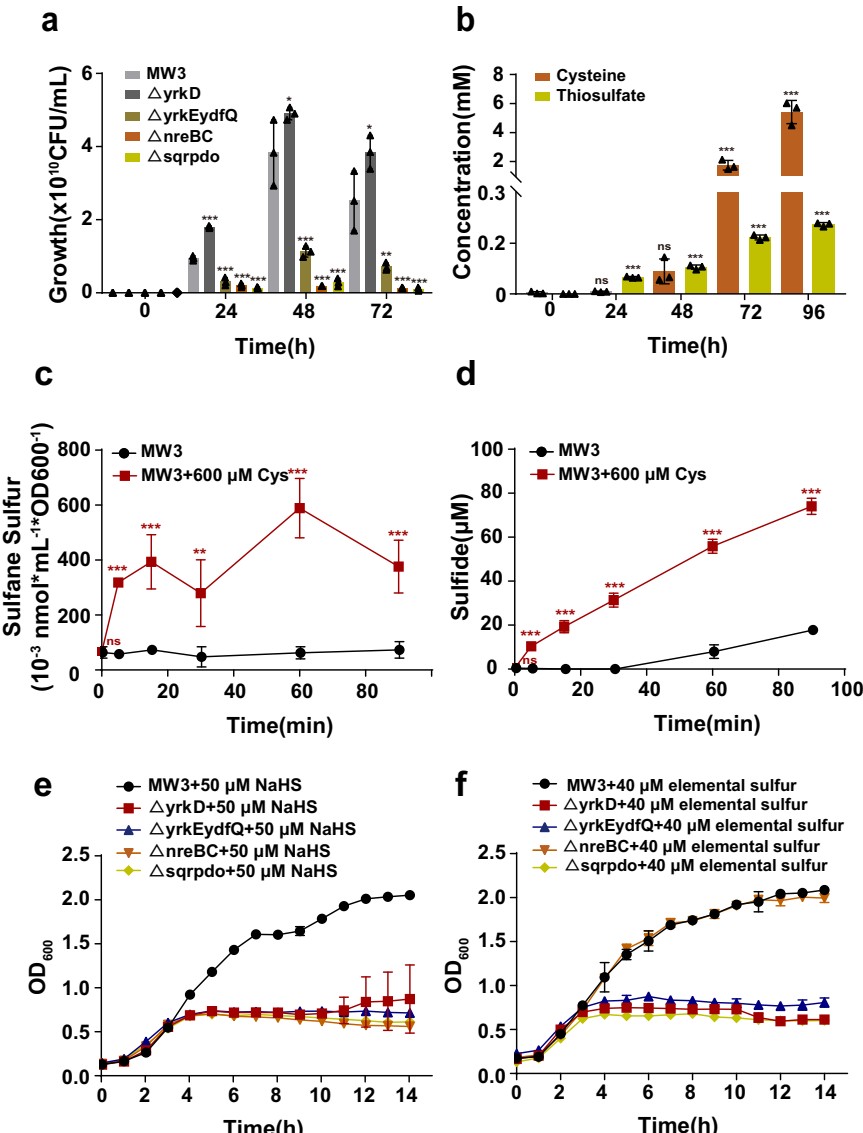

**Fig. 2 The *sqr*-containing gene cluster helped *B. licheniformis* to resist cysteine toxicity. a** The growth of *B. licheniformis* MW3 and its mutants in the feather medium. The initial cell counts were at $4.0 \times 10^4$ CFU mL$^{-1}$ approximately. **b** The identification of sulfur metabolites when *B. licheniformis* MW3 was inoculated into the feather medium. **c, d** *B. licheniformis* MW3 cells growing in LB were prepared into resting cells in HEPES buffer at an OD$_{600}$ of 2. 600 μM cysteine was added to start the test. The produced sulfide and intracellular sulfane sulfur were measured. *B. licheniformis* MW3 and its mutants were inoculated into an LB medium with or without 50 μM NaHS (**e**) and 40 μM elemental sulfur (**f**) at the initial OD$_{600}$ of 0.1 and cultured in a 24-well culture plate with continuous shaking at 37 °C. The OD$_{600}$ values were recorded at defined time intervals. Three parallel experiments were performed to obtain the averages and standard deviations for each figure. Data are presented as mean ± SD. One-way ANOVA was performed to calculate the *p*-values. For Fig. 2a, MW3 at each time point was used as the control. For Fig. 2b, the concentration at time 0 was used as control. For Fig. 2c, d, statistical comparisons are between MW3 and MW3 + Cys at each time point. Asterisks indicate statistically significant difference (*$p < 0.05$, **$p < 0.01$, ***$p < 0.001$, ns = ($p > 0.05$)).

converts cysteine to sulfane sulfur and sulfide, which were more toxic to the mutants than to the wild type.

**YrkD and NreBC synergistically controlled the expression of sqr.** *B. licheniformis* MW3 and its mutants were cultured in a modified M9 medium with or without sulfide or elemental sulfur for induction. The cells were harvested by centrifugation, washed once in 50 mM HEPES buffer (pH 7.0) containing 50 μM DPTA, 0.5 g/L NH$_4$Cl, and 20 g/L glucose, and resuspended in the same buffer at an OD$_{600}$ of 2.0. Sulfide oxidation by the wild type required elemental sulfur induction, and sulfide had a minimal effect as the inducer (Fig. 3a). Sulfide oxidation by the *yrkD* deletion mutant (Δ*yrkD*) required the induction of sulfide instead of elemental sulfur (Fig. 3b). The double deletion mutant of *nreB* and *nreC* (Δ*nreBC*) significantly lost its ability for sulfide oxidation, and either elemental sulfur or sulfide induction did not enhance sulfide oxidation (Fig. 3c). The *sqr* deletion mutant (Δ*sqr*) had marginal activity of sulfide oxidation (Fig. 3d). The *yrkD* gene and the *nreBC* genes under control of constitutive promoters were inserted into the downstream region of *pdo* gene in Δ*yrkD* mutant and Δ*nreBC* mutant, respectively. After complementation, Δ*yrkD::yrkD* was induced by elemental sulfur for sulfide oxidation (Fig. 3e). Δ*nreBC::nreBC* resumed the ability for sulfide oxidation (Fig. 3f). Since *nreB* and *nreC* were constitutively expressed in Δ*nreBC::nreBC*, both elemental sulfur and sulfide strongly induced sulfide oxidation with sulfide being stronger than elemental sulfur (Fig. 3f). These results suggest that YrkD represses the expression of *nreB* and *nreC*, and the repression is released after elemental sulfur induction. The NreBC system activates the transcription of *sqr* in the presence of sulfide.

**The 11 genes were co-transcribed.** The co-transcription of genes within the *yrkD-pdo* gene cluster was analyzed by using 10 pairs of primers targeting the 10 transgenic regions with the genomic DNA (gDNA), total RNA, and cDNA extracted from the *B. licheniformis* MW3 cells induced with elemental sulfur in modified M9 medium as the templates (Fig. 1a). PCR reactions with these primer pairs amplified the predicted products from the cDNA and gDNA (Fig. 4a and Supplementary Fig. 4a), but not from the total RNA (Supplementary Fig. 4b). The results indicate that the 11 genes are co-transcribed as a single polycistronic mRNA.

**Three promoters were present within the gene cluster.** Potential promoters were further screened by individually inserting the 10 intergenic regions before *egfp* in a promoter-screening plasmid (Fig. 4b). The IR1, IR2, and IR9 allowed the expression of *egfp* in *E. coli* (Fig. 4c). The positions of the three promoters were further analyzed by truncating the respective intergenic regions from its 5' direction via 60-bp walking (Supplementary Fig. 5 and Fig. 4d). The candidate promoters in these truncated intergenic regions were then predicted by using the Neural Network Promoter Prediction webserver (https://fruitfly.org/seq_tools/promoter.html) (Fig. 4e). The transcription initiation sites (TIS) of these three promoters were identified by using 5'-RACE from both the native strain and the recombinant *E. coli* strains (Fig. 4e). The TIS in IR9 was identified in both *B. licheniformis* MW3 and the corresponding recombinant *E. coli* strain. The TIS in IR1 was identified in *B. licheniformis* MW3, but not from the corresponding recombinant *E. coli* strain. The TIS upstream *yrkE* was identified from the corresponding recombinant *E. coli* strain (Fig. 4e). We tried to find the transcription start site of the *yrkE* promoter in *B. licheniformis* MW3, but no confident results were obtained. It was likely that the transcription from the promoter

was weak. The transcription from the *yrkD* promoter might cause the leaky expression of genes in the cluster besides *yrkD*.

To confirm that the promoter in IR9 works independently, *sqr* and *pdo* was reversed to generate an *sqr-pdo*-R strain (Fig. 5a), in which sulfide oxidation was marginally induced by sulfide but strongly induced by elemental sulfur (Fig. 5b), similar to the responses by MW3 (Fig. 3a). The results support that the promoter in IR9 is specific for the transcription of *sqr* and *pdo*. Without induction, MW3 had some activity of sulfide oxidation (Fig. 3a), but the *sqr-pdo*-R strain lost the activity (Fig. 5b). The results suggest that a basal level expression of *sqr* and *pdo* in MW3 but not in the *sqr-pdo*-R strain.

**YrkD sensed sulfane sulfur and NreB responded to sulfide.** The *egfp* gene was inserted behind the *pdo* gene on the chromosome of *B. licheniformis* MW3 and the Δ*yrkD* mutant to create *B. licheniformis* A1109 and A1120, respectively (Fig. 6a). The expression of *egfp* in *B. licheniformis* A1109 and A1120 growing in the modified M9 medium was induced with 100 μM different compounds for 90 min. In *B. licheniformis* A1109 with *yrkD*, elemental sulfur and polysulfide (H$_2$S$_n$) induced the production of eGFP in these cells, but NaHS, GSSG, sulfite, thiosulfate, and H$_2$O$_2$ did not (Fig. 6b). When different amounts of NaHS and elemental sulfur were used for the induction, elemental sulfur was more effective for the induction and the induction was increased as the inducer concentrations increased to 200 μM (Fig. 6c). The weak induction by NaHS is likely due the presence of small amounts of polysulfide in the NaHS stock solution[38].

*B. licheniformis* A1120 without *yrkD* had basal expression of *egfp* (Fig. 6d). NaHS, H$_2$S$_n$, and elemental sulfur further induced eGFP fluorescence, but GSSG, sulfite, thiosulfate, and H$_2$O$_2$ did not (Fig. 6d). When different amounts of NaHS and elemental sulfur were used for the induction, both reached the maximal induction at around 50 μM and NaHS was more effective than did elemental sulfur (Fig. 6e). When A1120 was exposed to 5, 10, and 20 μM sulfide or elemental sulfur, the transcript of *sqr* was determined by using RT-qPCR. The *sqr* transcription responded to sulfide, but not to the low levels of elemental sulfur after 5-min induction (Fig. 6f). The low level of induction of sulfide oxidation by increased elemental sulfur in *B. licheniformis* A1120 is likely due to the reduction of elemental sulfur to H$_2$S by small cellular thiols[37].

**The crosstalk of YrkD and NreBC regulated sqr and pdo.** The *yrkD* gene was deleted from the *sqr-pdo*-R mutant to generate M1109 (Fig. 7a). MW3 and M1109 were cultured in the modified M9 medium, induced by adding 100 μM elemental sulfur or NaHS, and analyzed by using RT-qPCR at 2, 10, 30 min of the induction. MW3 was induced with elemental sulfur, as sulfide was not effective as an inducer (Figs. 3a and 6b). In MW3 induced with elemental sulfur, the expressions of *yrkD*, *yrkH*, and *nreC* were the highest before 10 min, but the expression of *sqr* was the highest at 30 min (Fig. 7b–e). A1109 was induced with sulfide, as strains with the *yrkD* deletion did not respond to elemental sulfur (Fig. 6f). In M1109, *yrkH*, and *nreC* were not induced by NaHS (Fig. 7c, d); however, *sqr* was induced by NaHS, and its expression reached the highest level at 10 min with the change being significantly more than that in MW3 induced by elemental sulfur (Fig. 7e). To confirm that the NreBC system was solely responsible for the sulfide-dependent induction of the promoter in IR9, the *nreBC* gene was deleted from the M1109 strain to generate M1109-Δ*nreBC* (Supplementary Fig. 6a), and the transcription of *sqr* was abolished in M1109-Δ*nreBC* (Supplementary Fig. 6b). These results suggest that YrkD controls the expression

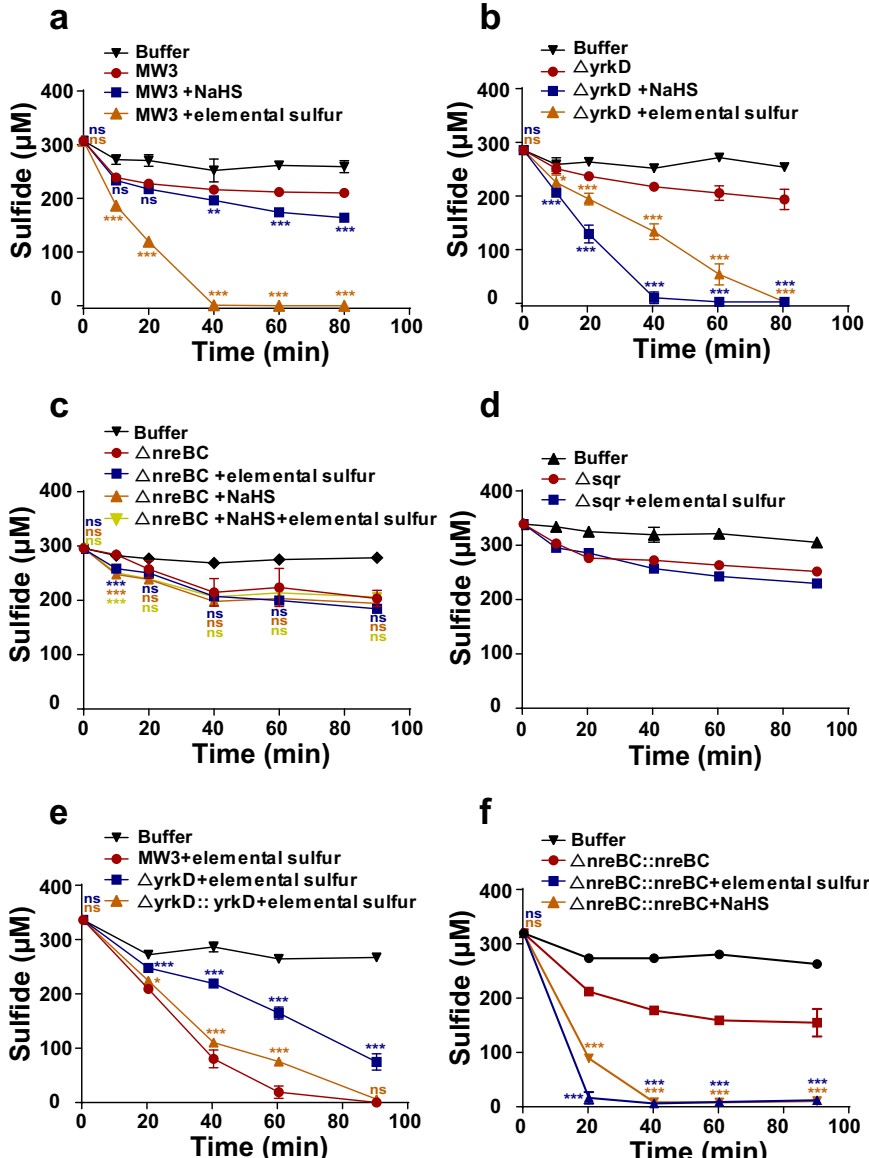

**Fig. 3 The effect of gene deletions on sulfide oxidation by *B. licheniformis*.** *B. licheniformis* MW3 and its mutants were cultured in the modified M9 medium with or without induction by 200 µM sulfide, or 200 µM elemental sulfur, or both for 2 h before harvesting to prepare resting cells in HEPES buffer at $OD_{600}$ of 2.0. The resting cells were used to oxidize 400 µM sulfide. **a** The sulfide oxidation curves for MW3; **b** The sulfide oxidation curves for $\Delta yrkD$; **c** The sulfide oxidation curves for $\Delta nreBC$; **d** The sulfide oxidation curves for $\Delta sqr$; **e** The sulfide oxidation curves for $\Delta yrkD::yrkD$ with MW3 and $\Delta yrkD$ as controls; **f** The sulfide oxidation curves for $\Delta nreBC::nreBC$ with MW3 and $\Delta nreBC$ as controls. "+NaHS" and "+elemental sulfur" shown in the labels mean the indicated strain was induced by sulfide and elemental sulfur, respectively. Three parallel experiments were performed to obtain the averages and standard deviations. Data are presented as mean ± SD. One-way ANOVA was performed to calculate the *p*-values. Asterisks indicate statistically significant difference (*$p < 0.05$, **$p < 0.01$, ***$p < 0.001$, ns = ($p > 0.05$)). For Fig. 3a–d, asterisks indicate statistical comparisons to the group of strains without induction at a given time point. For Fig. 3e, asterisks indicate statistical comparisons to the group of MW3 + elemental sulfur at a given time point. For Fig. 3f, asterisks indicate statistical comparisons to the group of $\Delta nreBC::nreBC$ without induction at a given time point.

of all the 11 genes in this gene cluster (Fig. 1a), and the NreBC system activates *sqr* and *pdo*.

When 10 µM to 500 µM elemental sulfur was used to induce MW3 for 10 min, the expression of *yrkD, yrkH, nreC,* and *sqr* reached the highest level at 500 µM, 100 µM, 50 µM, and 100 µM, respectively (Fig. 8a–d). Notably, *yrkH* had a 5.3-fold higher transcriptional change than that of *nreC*. A hairpin structure in the intergenic region between *yrkI* and *yrkJ* may affect the expression strength of *nreC* (Fig. 8e). The *sqr* expression had the least response to elemental sulfur (Fig. 8d).

To test whether the presence of NreBC was an indispensable factor to initiate the transcription of *sqr*, the transcription level of

*sqr* in MW3, $\Delta yrkD$, and $\Delta yrkD+\Delta nreBC$ mutants were measured after culturing in the modified M9 medium without induction. The *sqr* transcript was hardly detectable in the uninduced MW3 and was increased 87.0-fold in the $\Delta yrkD$ mutant and only 25.6-fold in the $\Delta yrkD+\Delta nreBC$ mutant (Fig. 8f). The results support that YrkD inhibits the transcription of all the genes in the cluster, including *sqr* and *pdo*, and the NreBC system further amplifies the expression of *sqr* and *pdo*.

**YrkD and NreC bound to their cognate binding sites.** The recombinant YrkD with a C-terminal His-tag and NreC with

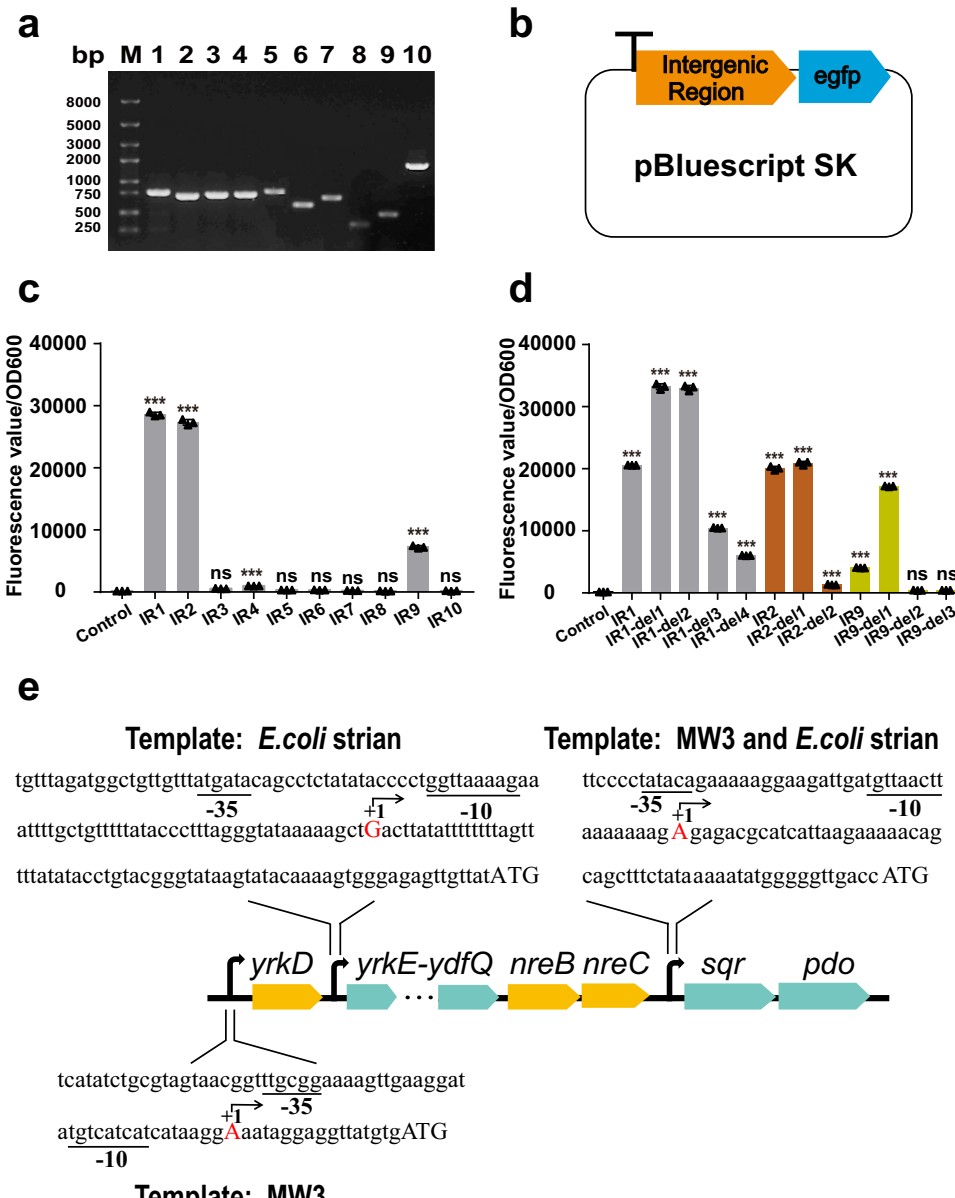

**Fig. 4 The promoters and the transcription initiation sites of the gene cluster. a** The reverse-transcription PCR results with the 10 pairs of primers to amplify the ten intergenic regions (Fig. 1a) from cDNA of the induced *B. licheniformis* MW3. **b** The pBluescript SK- plasmid contained *egfp* as the reporter gene. A terminator was inserted in the front of the *egfp* gene to prevent potential transcription of *egfp* from an upstream promoter. The 10 intergenic regions (IR1-IR10, Fig. 1a) were assembled between the terminator and the *egfp* gene with no intergenic region group insertion as control. **c** The fluorescence intensities of the reporter screening system. A promoter-reporter plasmid without any intergenic region was used as the control. *E. coli* strain MG1655 was used as the host. **d** The intergenic regions of IR1, IR2, and IR9 were further truncated from 5'-end. The detailed sequence information was shown in Supplementary Fig. 5. The initiation strength of the truncated promoter was assayed with the eGFP system. **e** The annotated promoters with the transcription initiation sites. When appropriate, three parallel experiments were performed to obtain the averages and standard deviations. Data are presented as mean ± SD. One-way ANOVA was performed to calculate the *p*-values with the no intergenic region group insertion group as control, and asterisks indicate statistically significant difference (*$p < 0.05$, **$p < 0.01$, ***$p < 0.001$, ns = ($p > 0.05$)).

N-terminal maltose-binding protein (MBP) was purified (Supplementary Fig. 7a, b). The binding sites of YrkD and NreC were screened from the intergenic sequences harboring the promoters. YrkD bound to IR1 and IR2 (Fig. 9a, b), but not to IR9 (Supplementary Fig. 8). NreC could bound to IR9, but the binding band is not clear. When NreC was phosphorylated by acetyl-phosphate, the phosphorylated NreC had a higher affinity to IR9 and could form clear binding bands (Supplementary Fig. 9a, Fig. 9c, d), but not bound to IR1 and IR2 (Supplementary Fig. 9b). The IR1 and IR2 were further divided into 7 segments

and 2 fragments, respectively (Supplementary Fig. 10). YrkD specifically bound to both the IR1–6 fragment and the IR2-2 fragment (Fig. 9e–g). Elemental sulfur-treated YrkD did not bind to these two fragments (Fig. 9h). Because the DNA binding domains of CstR$_{Sa}$ from *S. aureus* and NreC$_{Sc}$ from *Staphylococcus carnosus* had 50% and 40% sequence identities to those of YrkD and NreC from *B. licheniformis* MW3, their DNA binding sites were used to predict the potential binding sites of YrkD and NreC (Supplementary Fig. 11a, b). When the predicted binding sites were mutated (Supplementary Fig. 11a, b), YrkD and

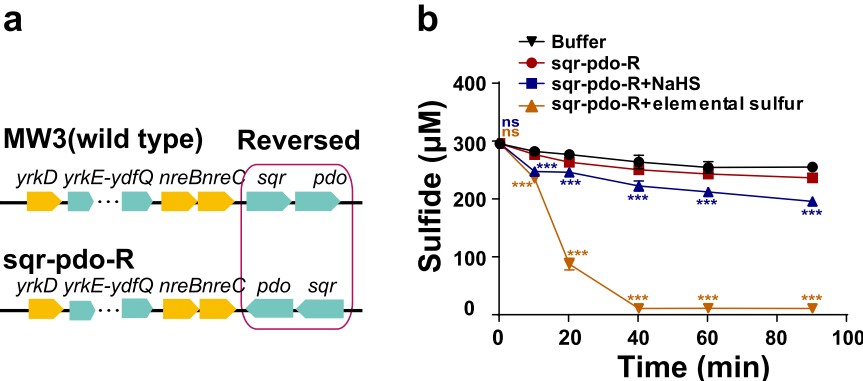

**Fig. 5 The confirmation of the promoter between *nreC* and *sqr*. a** The schematic diagram of the *sqr-pdo*-R strain. **b** The *sqr-pdo*-R strain was cultured in the modified M9 medium with the induction of 200 μM sulfide or 200 μM elemental sulfur for 2 h. The resting cells were used to oxidize sulfide. Three parallel experiments were performed to obtain the averages and standard deviations. Data are presented as mean ± SD. One-way ANOVA was performed to calculate the *p*-values. Asterisks indicate statistically significant difference (\*\*\**p* < 0.001, ns = (*p* > 0.05)), and asterisks indicate statistical comparisons to the *sqr-pdo*-R at a given time point.

phosphorylated NreC lost binding to the mutated DNA probes (Fig. 9i, j). The $K_d$ values of YrkD to its binding sites in IR1 and IR2 were determined by using fluorescence polarization as 2.4 μM and 2.3 μM, respectively (Supplementary Fig. 12a, b). The $K_d$ values of the elemental sulfur-treated YrkD were too high to be determined (Supplementary Fig. 12a, b). The $K_d$ value of NreC to its binding site was at 59.0 μM, and the phosphorylation of NreC lowered it to 9.8 μM (Supplementary Fig. 12c, d).

## Discussion

*B. licheniformis* MW3 can oxidize self-produced and exogenous $H_2S$ via an SQR-PDO pathway, which is common in heterotrophic bacteria[10]. The sulfur-oxidizing genes are located in a gene cluster of 11 genes (Fig. 10), encoding three regulatory proteins YrkD, NreB, and NreC. YrkD represses two promoters: one starts before its gene *yrkD* to give a transcript of all the 11 genes, and the other starts after *yrkD* leading to a transcript of the rest ten genes. The presence of two promoters could enhance the transcription intensity of the latter ten genes (Fig. 10). The low-level expression of *sqr* and *pdo* in the Δ*yrkD* mutant and the detection of the co-transcription of *nreC* and *sqr* support that *sqr* and *pdo* are also transcribed from the more distal promoters (Figs. 4a, 6f, and 7e). The reduced levels of *sqr* and *pdo* transcripts in comparison with those of other genes in the gene cluster could be caused by selective RNA processing and stabilization, which breaks a polycistronic mRNA into small fragments and their stability is determined by secondary structures at the 3'-ends of each fragment[39]. However, the expression of *sqr* and *pdo* from the two promoters regulated by YrkD is not sufficient, as the deletion of *nreB* and *nreC* essentially abolishes sulfide oxidation (Fig. 3c), and the induction profile of *sqr* and *pdo* in the *sqr-pdo*-R strain was almost the same as that of MW3 (Figs. 3a, 5b). Their adequate expression requires the NreBC system. The NreBC system is activated by sulfide, which greatly increases the transcription level of *sqr* and *pdo* (Fig. 7e).

Among the four reported regulatory proteins responding to sulfur oxidation[24–26,40,41], CstR is a homolog of YrkD with ~50% sequence identity (Fig. 1a). CstR uses two conserved cysteine residues to sense sulfane sulfur and forms di- and trisulfide bridges between two cysteine residues in the adjacent monomers of the tetramer, and the modified CstR derepresses its controlled promoters[26,28]. YrkD also contains the two conserved cysteine residues. Hence, YrkD is expected to function similarly as CstR does.

NreBC is the homolog of NreBC (NreBC$_{Sc}$) from *S. carnosus*[42,43]. Our result has shown that both NreC proteins have 40.0% sequence identity and recognize similar DNA-binding sites (Fig. 9j, Supplementary Fig. 11b). However, the two systems have differences. The NreBC$_{Sc}$ system contains an additional component NreA$_{Sc}$[44,45]. NreA$_{Sc}$ is a nitrate receptor, and the nitrate-bound NreA$_{Sc}$ interacts with NreB and inhibits its phosphorylation[44]. The sequence identity between the PAS domains of the two NreB proteins is low at 26.7%. Its phylogeny relation with known homologs indicate that the PAS domain of NreB is different from characterized homologs (Fig. 1c). All these representative homologs of PAS domain that use heme B as the cofactor sense $O_2$, CO, or NO[46], but the PAS domain of NreB senses $H_2S$ (Fig. 6e, f). These results suggest that the PAS domain of NreB has different structural features to sense $H_2S$.

NreB may use its bound heme to sense sulfide, as two heme containing proteins have been reported to sense $H_2S$. The *E. coli* direct oxygen sensor (DosP) contains two domains, a PAS domain with heme and a catalytic domain that hydrolyzes bis-(3′-5′)-cyclic guanosine monophosphate (c-di-GMP). The reduced form ($Fe^{2+}$) or the oxidized form ($Fe^{3+}$) of DosP is not very active, the activities of the $O^{2-}$ complexed the reduced DosP ($Fe^{2+}$-$O_2$) or the sulfide complexed the oxidized DosP ($Fe^{3+}$-$SH^-$) are markedly increased. The oxidized ($Fe^{3+}$) DosP senses sulfide at relatively high concentrations from 50 to 500 μM[47]. Recently, the direct oxygen sensor DosS from *Mycobacterium tuberculosis* has been shown to sense undissociated $H_2S$[48]. DosS contains a GAF domain that contains a heme, and the oxidized form with heme $Fe^{3+}$ has low kinase activity. When it binds undissociated $H_2S$, $Fe^{3+}$ is reduced to $Fe^{2+}$, which activates its kinase activity. Then, DosS can transfer the phosphoryl group to DosR that regulates the dormancy regulon in *M. tuberculosis*. The PAS domains and the GAF domains are structurally similar[49]. However, the PAS domain of NreB was not phylogenetically related to those of DosP and DosS (Fig. 1c). How NreB senses sulfide requires further investigation.

In bacteria, cascade regulations are not uncommon, and they regulate different functions, such as flagella, quorum sensing, and virulence[50–52]. Most cascade regulations are initiated by a global transcription factor that activates other regulatory factors, and the signals of these regulatory factors are normally not the same. Here, a sulfide oxidation pathway is regulated by a cascade regulation in *B. licheniformis* MW3. The signals are $H_2S$ and sulfane sulfur, which are mutually converted[22,53]. $H_2S$ is oxidized by SQR to sulfane sulfur, and sulfane sulfur is reduced by cellular thiols to form $H_2S$ and oxidized thiols[54,55]. *B. licheniformis* MW3 uses

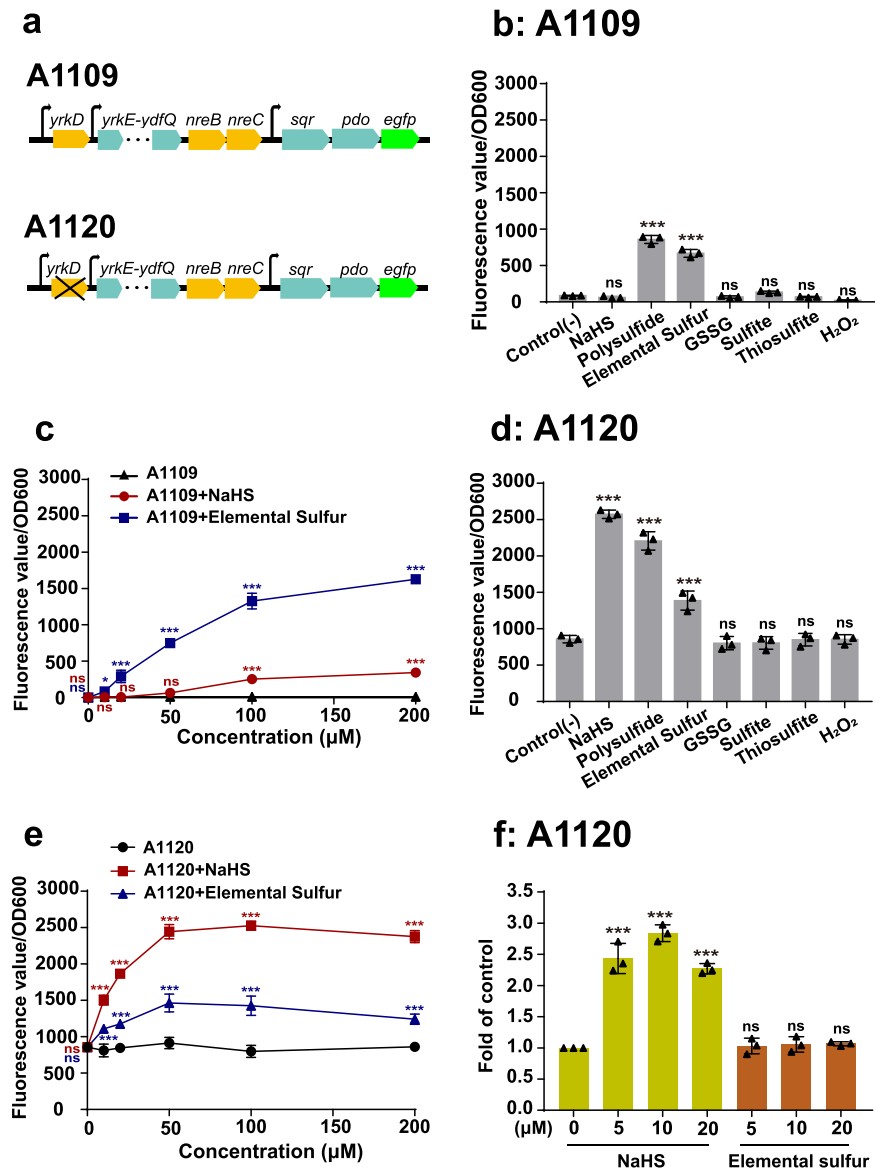

**Fig. 6 The inducers of YrkD and NreBC. a** The schematic graph for the insertion of *egfp* behind *pdo* in *B. licheniformis* MW3 and its mutant Δ*yrkD* to generate A1109 and A1120. **b** 100 μM potential inducers were used to induce eGFP expression in A1109 for 90 min. **c** Gradient concentrations of NaHS and elemental sulfur were used to induce the expression of eGFP in A1109 for 90 min; **d** 100 μM potential inducers were used to induce eGFP expression in A1120 for 90 min. **e** Gradient concentrations of NaHS and elemental sulfur to induce the expression of eGFP in A1120 strain for 90 min. **f** The fold changes of *sqr* transcription in A1120 after induction with 5 μM, 10 μM, and 20 μM of sulfide (green) and elemental sulfur (brown) for 5 min. Three parallel experiments were performed to obtain the averages and standard deviations. Data are presented as mean ± SD. One-way ANOVA was performed to calculate the *p*-values with the uninduced group as control, and asterisks indicate statistically significant difference (*$p < 0.05$, ***$p < 0.001$, ns = ($p > 0.05$)).

bacillithiol as its main intracellular low-molecular-weight thiol[56], and it has similar chemical properties as GSH[57]. Hence, sulfide could be produced by oxidizing bacillithiol if sulfane sulfur accumulates in cells. The transcription of *sqr* and *pdo* is affected by several factors. First, YrkD is inactivated by sulfane sulfur, which leads to a low-level transcription of *sqr* and *pdo*, and the produced SQR may further increase cellular sulfane sulfur (Figs. 7e and 8f). The increased NreB and NreC initiate the transcription of *sqr* and *pdo* at their promoter (Fig. 8f). When NreB senses H₂S, the transcription of *sqr* and *pdo* is significantly amplified (Fig. 7e).

The gene cluster also contains other genes related to sulfur metabolism besides *sqr*, *pdo*, and the three genes encoding the two regulatory systems (Fig. 10). A *yrkE-ydfQ* gene cluster appears with an unknown function. YrkH is likely a second PDO,

as it is homologous to other PDOs. TauE is a potential sulfite transporter, and its genes are commonly associated with *sqr* and *pdo* gene clusters[10,25,26,36,58]. YdfQ is homologous to thioredoxins. Thioredoxin reduces persulfide to H₂S[59]. YrkE, YrkF, and YrkI are homologs to DsrE, RHOD, and TusA, respectively (Fig. 10). In the purple sulfur bacterium *Allochromatium vinosum*, the sulfane sulfur in the form of a persulfide is transferred from RHOD to TusA, and then to DsrE; the final acceptor of the sulfane sulfur is DsrC, and DsrAB oxidizes the sulfane sulfur in DsrC to sulfite[60,61]. In *S. aureus*, the three genes are fused into *cstA* within the gene cluster containing *sqr* and *pdo*[34]. The RHOD domain of CstA abstracts the sulfane sulfur from thiosulfate and forms a persulfide at its active site. The sulfane sulfur of the persulfide on the RHOD domain can be transferred to a cysteine

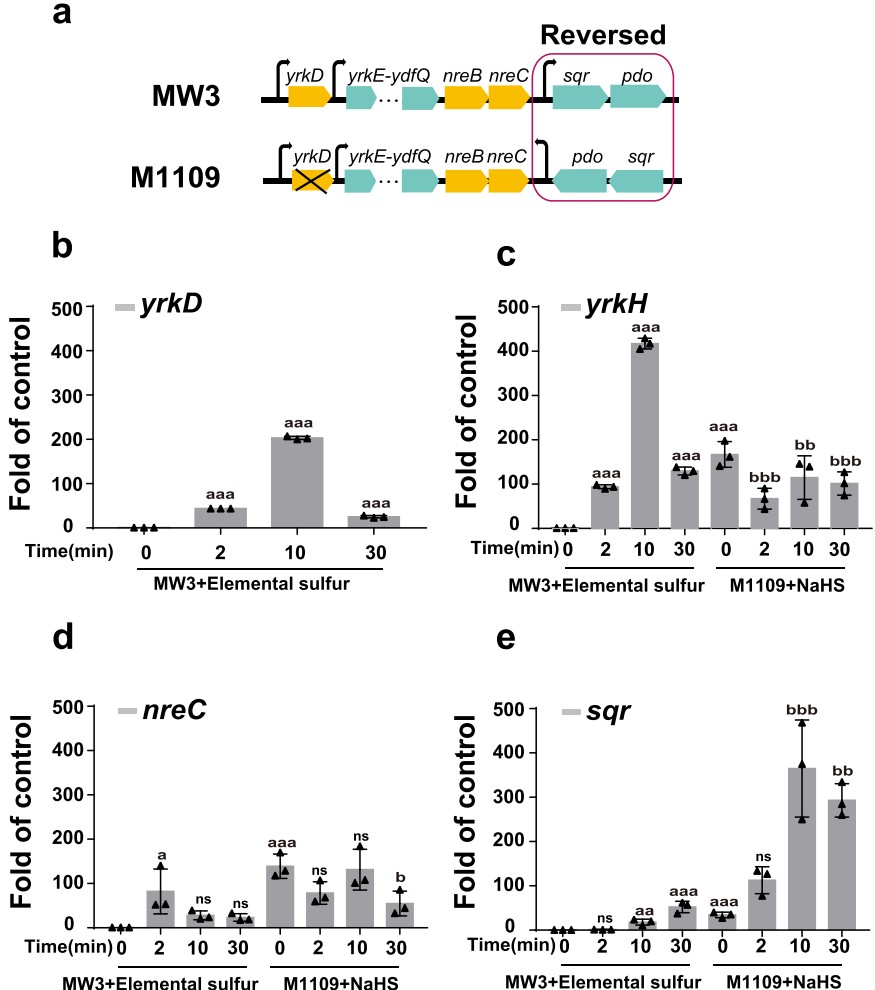

**Fig. 7 The transcription level of indicated genes of the gene cluster of WM3 and a *yrkD* deletion mutant M1109 with inverted *sqr* and *pdo*. a** The schematic graph for the conversion of MW3 to M1109. **b**–**e** MW3 and M1109 were cultured in the modified M9 medium. The MW3 was induced with 100 μM elemental sulfur, and the M1109 was induced with 200 μM NaHS. After induction, the changes in transcription levels of *yrkD*, *yrkH*, *nreC*, and *sqr* were measured. For Fig. 7c–e, The first four time points indicated MW3 with induction by elemental sulfur, and the last four time points indicated M1109 with induction by NaHS. Three parallel experiments were performed to obtain the averages and standard deviations. Data are presented as mean ± SD. One-way ANOVA was performed to calculate the *p*-values, and "a" indicates the control used for comparison is the MW3 without induction, and "b" indicates the control used for comparison is the M1109 without induction. ([a/b]$p < 0.05$, [aa/bb]$p < 0.01$, [aaa/bbb]$p < 0.001$, ns = ($p > 0.05$)).

residue of the TusA domain, suggesting these domains are the sulfurtransferase and carrier. However, the physiological functions of YrkE, YrkF, YrkI, and YdfQ in *B. licheniformis* are unknown. Hence, these 6 genes encode proteins that are related to sulfane sulfur transfer, oxidation, or reduction, but their physiological functions during H₂S oxidation in *B. licheniformis* warrant further investigation.

One physiological function of this gene cluster is to help *B. licheniformis* cells to adapt the feather habitat (Fig. 2a). Feather keratin is rich in cysteine residues, and our results suggest that *B. licheniformis* degrades feather into cysteine (Fig. 2b). The accumulation of cysteine acts as a double-edged sword. A recent report suggests that reduced cysteine participates in the reduction of disulfide bonds of keratin protein[6]; however, extra cysteine is toxic to the bacterial cells (Supplementary Fig. 2a). This gene cluster helps the bacterial cells to resist cysteine stress, as the mutants defective in the expression of *sqr-pdo* and *yrkE-ydfQ* are more sensitive to added cysteine (Supplementary Fig. 2b–d). Exogenous cysteine greatly increases cellular sulfane sulfur and H₂S in *B. licheniformis* (Fig. 2c, d). In turn, sulfane sulfur and H₂S induce the production of SQR and PDO that can oxidize them

and lower their cellular concentrations (Fig. 3). Both YrkD and NreBC were necessary for the cells to resist the sulfide stress, and only YrkD was necessary for the cell to deal with the sulfane sulfur stress (Fig. 2e, f). The results indicate that the cascade regulation has a division of labor. The regulation by sulfane-sulfur sensing YrkD is critical against sulfane sulfur toxicity, and the presence of SQR and PDO controlled by NreBC was more special to deal with sulfide stress. The cascade regulation of this pathway is also important to resist cysteine stress. Puzzlingly, the deletion of YrkD was not good for cells to resist high cysteine concentration on an agar plate (Supplementary Fig. 2c, d), although MW3 grew better than ΔyrkD did in the presence of 50 μM sulfide (Fig. 2e). Possibly, the increased basal level expression of *sqr* may convert H₂S to sulfane sulfur, and the lowered level of H₂S reduces the expression of *sqr* and *pdo* by the NreBC system. Thus, the cascade regulation of this gene cluster helped the bacterium deal with cysteine toxicity.

In summary, we identified a gene cluster involved in sulfur oxidation from the feather-degrading *B. licheniformis* MW3 (Fig. 10). The sulfur oxidation participates in the detoxification of cysteine, which is one of the main amino acids for feather keratin.

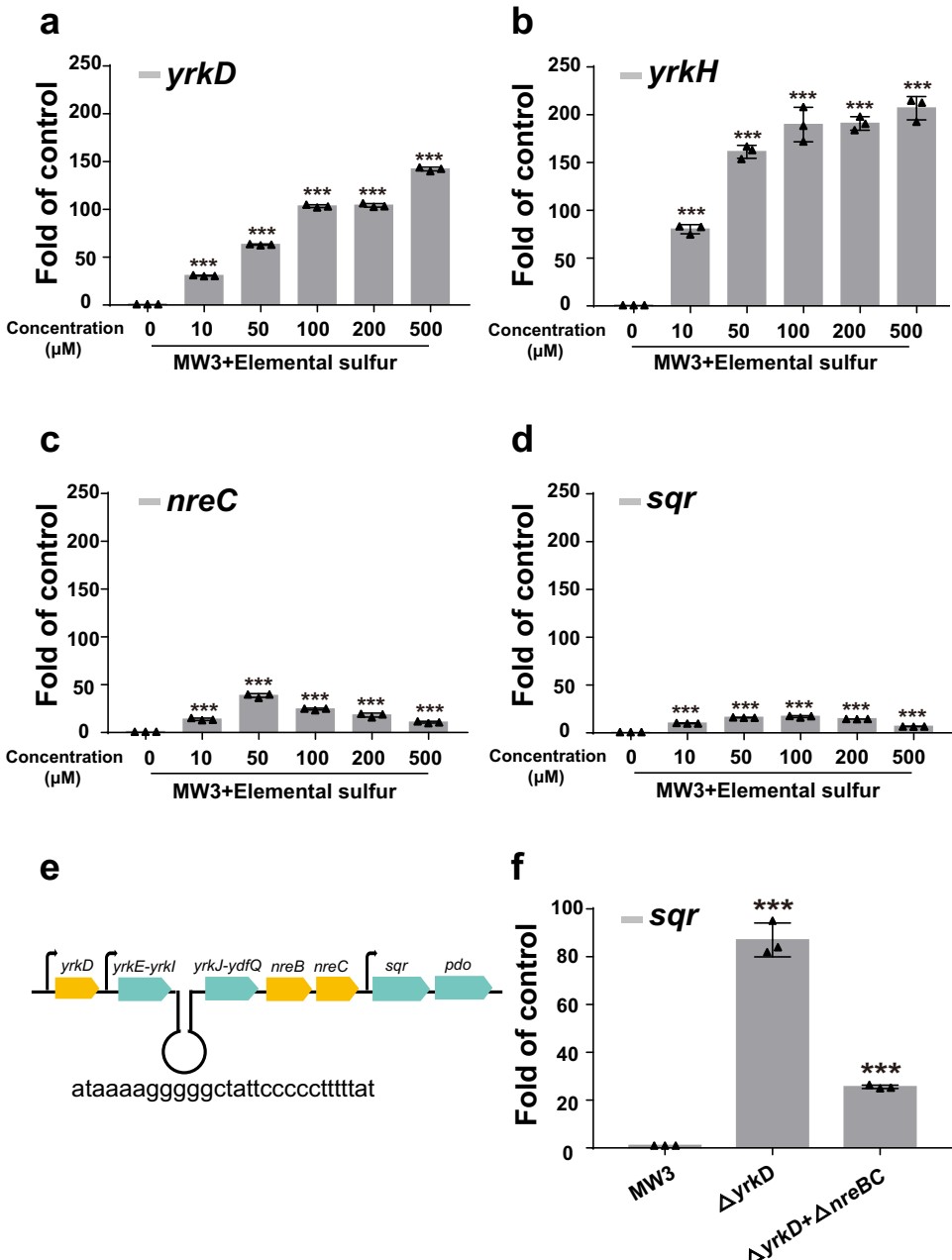

**Fig. 8 The transcription level of indicated genes in MW3 and its mutants (ΔyrkD and ΔyrkDΔnreBC). a–d** MW3 was cultured in the modified M9 medium. The MW3 was induced with a gradient of elemental sulfur. MW3 without induction was used as the control. After induction for 10 min, the transcription levels of *yrkD*, *yrkH*, *nreC*, and *sqr* were measured by using the RT-qPCR method; **e** A schematic map of gene cluster to show the hairpin structure between *yrkI* and *yrkJ*; **f** The three indicated strains were cultured in the modified M9 medium without induction. The transcription levels of *sqr* in these strains were measured by using the RT-qPCR method. The *gyrB* was used as a reference gene. Three parallel experiments were performed to obtain the averages and standard deviations. Data are presented as mean ± SD. One-way ANOVA was performed to calculate the *p*-values, and asterisks indicate a statistically significant difference (****p* < 0.001). For Fig. 8a–d, the uninduced group was used as the control for comparison. For Fig. 8f, the MW3 group was used as the control for comparison.

The genes were regulated by two regulators that displayed cascade regulation. The repressor YrkD was derepressed by sulfane sulfur to activate the production of NreB and NreC. NreB that sensed $H_2S$ to further enhance the expression of *sqr* and *pdo*. The cascade regulation of this pathway helped cells to adapt to the feather habitat by precisely resisting cysteine toxicity at different levels. Further, a type of sulfide sensor was identified in bacteria as far as we can tell. The findings may help with bioremediation of feather waste and sulfide pollution.

## Methods

**Bacterial strains, plasmids, culture media, and chemicals**. The bacterial strains and plasmids used in this study are listed in Supplementary Data 2. The used oligonucleotides and their usages are listed in Supplementary Data 3. *Escherichia coli* XL1-Blue MRF' was used as a cloning host and *E. coli* MFDpir was used as a donor host for conjugation. *E. coli* strains were cultured in lysogeny broth (LB). The *B. licheniformis* MW3 and its gene deletion mutants were cultured in an LB medium or a modified M9 medium at 37 °C. The modified M9 medium was prepared by mixing M9 medium with a 3.5% proportion of LB medium. The small amount of LB medium was necessary to provide the growth factor for the growth of *B. licheniformis* MW3 in the modified M9 medium (Supplementary Fig. 13). The *B.*

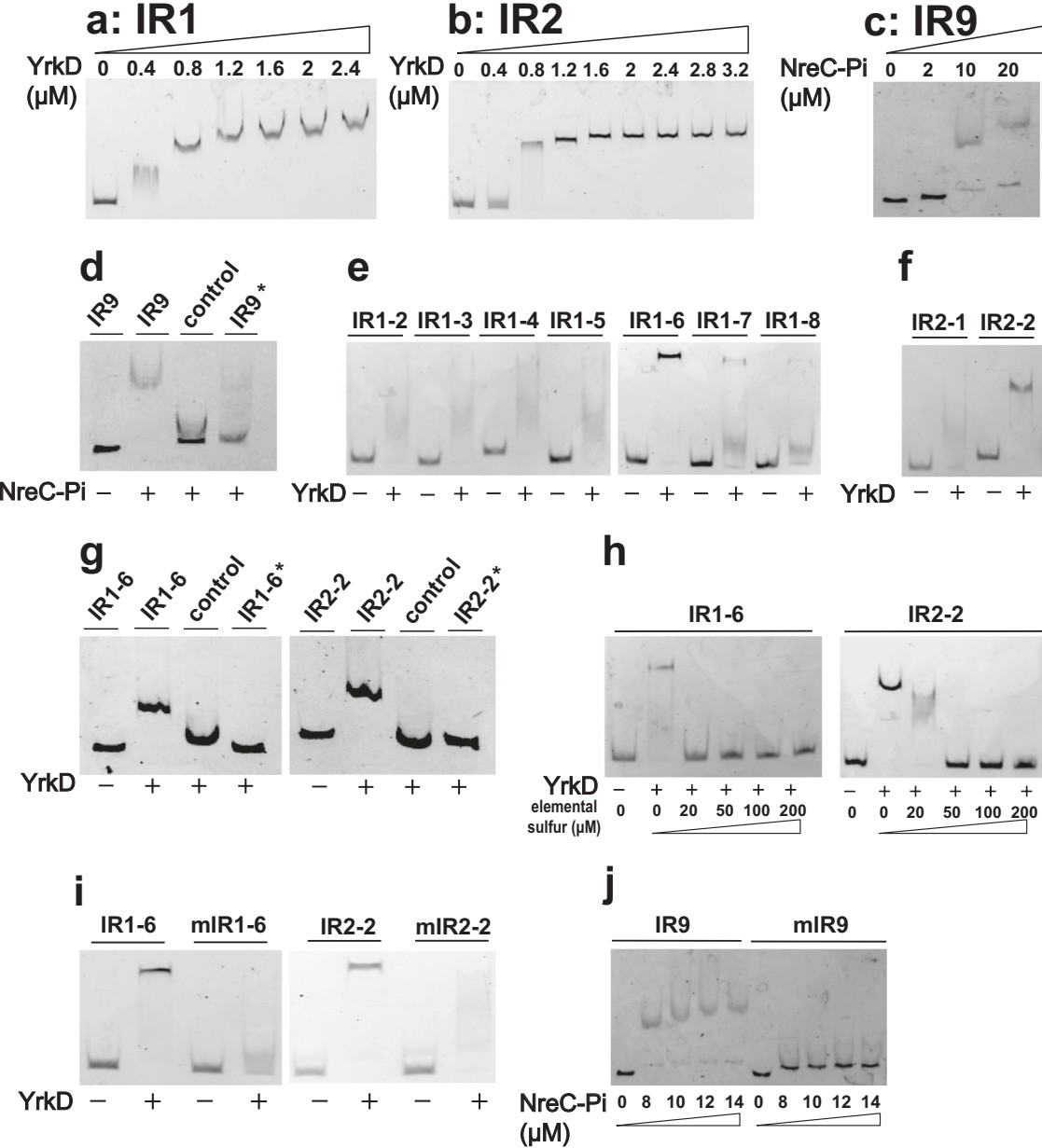

**Fig. 9 The DNA binding properties of YrkD and NreC. a–c** The intergenic region of IR1, IR2 and IR9 were amplified by using primers with 5'-FAM (5'-Carboxyfluorecein) labeling. The purified DNA fragments were mixed with purified YrkD and NreC, respectively. The binding ability was characterized by using EMSA. **a** The binding of YrkD to IR1. **b** The binding of YrkD to IR2. **c** The binding of NreC-Pi to IR9. **d** The binding specificity test for NreC-Pi to IR9. A fragment within *nreC* (147 bp) was used as the negative control for nonspecific binding. **e** EMSA of the binding of YrkD to the divided DNA fragments of IR1. **f** EMSA of the binding of YrkD to the divided DNA fragments of IR2. **g** The binding specificity tests for YrkD to IR1–6 or IR2-2. A fragment within *yrkD* (131 bp) was used as the negative control for nonspecific binding. **h** The addition of elemental sulfur blocked YrkD from binding to IR1–6 and IR2-2. In the reaction mixture, elemental sulfur was added to the indicated concentrations. **i** The binding of YrkD to mutated IR1–6 (mIR1–6) and IR2-2 (mIR2-2) was assayed with native IR1–6 and IR2-2 as the positive control. **j** The binding of NreC-Pi to mutated IR9 (mIR9) was assayed with native IR9 as the positive control. NreC-Pi means the NreC protein was phosphorylated before the EMSA test. The fragments marked with '*' indicated the addition of 600 nM FAM-free DNA as the cold competition control. In all binding specificity tests, 20 nM DNA probes were mixed with 2 µM YrkD or 10 µM NreC-Pi in addition to ~6.6 ng/µL poly(dI-dC), respectively.

*licheniformis* MW3 and its gene deletion mutants were also cultured in a feather medium. The feather medium was modified from a report method[62], and it contained 0.5 g/L NH$_4$Cl, 0.5 g/L NaCl, 0.3 g/L K$_2$HPO$_4$, 0.4 g/L KH$_2$PO$_4$, 0.24 g/L MgCl$_2$•6H$_2$O, 0.1 g/L yeast extract, 0.01 g/L CaCl$_2$, 0.24 g/L MgSO$_4$, 1% (w/v) chopped feather, pH 7.5. White secondary feathers were collected from a nearby poultry farm, cleaned thoroughly in distilled water, and dried at 60 °C in an oven. The feathers were chopped and added to the medium. Ampicillin (Amp) and kanamycin (Kan) were used at 100 µg/mL and 50 µg/mL, respectively, for *E. coli* strains. Erythromycin (Em) and polymyxin B were used at 5 µg/mL and 50 µg/mL, respectively, for *B. licheniformis* mutants for plasmid selection and maintenance.

X-Gal was added at 40 µg/mL for blue-white screening as mentioned[63]. The growth of MW3 and its mutants under different culture conditions was mainly monitored with optical density at 600 nm with a spectrophotometer (UV-1800, Shimadzu, Japan) except in the feather medium, where the method of counting colonies by diluting the bacterial cultures on fresh LB plate without antibiotics was employed.

Plasmid extraction, DNA purification, and genomic DNA extraction kits were purchased from TianGen (China), and enzymes were purchased from NEB (UK). X-Gal was purchased from Sangon Biotech (China). Other chemicals including sodium hydrosulfide (NaHS), sodium thiosulfate (Na$_2$S$_2$O$_3$), sodium sulfite (Na$_2$SO$_3$), and oxidized glutathione (GSSG) were purchased from Sigma-Aldrich

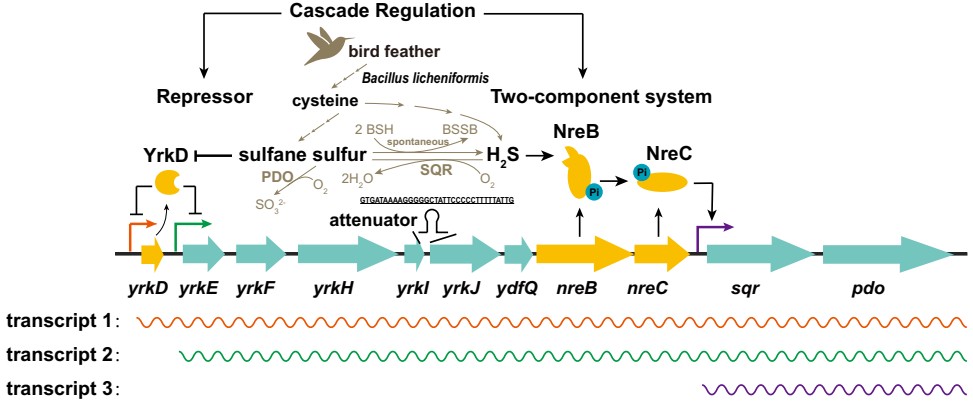

**Fig. 10 The schematic diagram for the cascade regulation of the sulfur-oxidizing gene cluster in *B. licheniformis* MW3.** The arrows on the black line indicate the genes. The regulatory genes are shown in yellow, and the rest are shown in cyan. The regulators are shown in different shapes in yellow. The solid bold black arrows indicate the activation of gene expression, and the line with a stop character means inactivation. A hairpin structure is present between *yrkI* and *yrkJ*. The possible metabolism relations among bird feathers, cysteine, sulfane sulfur, and sulfide were indicated with arrows in dark gold color. Consecutive arrows indicate that there may be a multi-step reaction. BSH reduced bacillithiol, BSSB oxidized bacillithiol, SQR sulfide: quinone reductase, PDO persulfide dioxygenase.

(US). Their concentrations were used as indicated in the text. Inorganic polysulfide was prepared by following a method as previously reported[15]. In brief, 0.5 mmol sulfur powder and 1.0 mmol sodium sulfide were added to 5 mL anoxic distilled water under argon gas, and the bottle was sealed. The bottle was incubated at 37 °C until sulfur was completely dissolved. The pH was adjusted to 9.3 with 6 M HCl, and the solution was diluted to 10 mL with anoxic distilled water.

**Construction of gene markerless deletion mutants, *egfp* insertion mutants and gene inversion mutant of *B. licheniformis* MW3.** The markerless deletion mutants of *B. licheniformis* MW3 were constructed using double-cross homologous recombination with the conjugative shuttle vector pKVM1[63]. Briefly, 1000-bp upstream and downstream homologous arms of each target gene were amplified by PCR and cloned into the *EcoRI* treated pKVM1 plasmid by using the TEDA method[64] with *E. coli* XL1-Blue MRF' as the host[64]. After conjugation and homologous recombination, the target gene was deleted in *B. licheniformis* MW3. To avoid the polar effect caused by gene deletion, sequences including 3 codons at the 5' end and 6 codons at the 3' end of the gene were kept. By using a similar way, *egfp* was cloned with amplified homologous arms from the *B. licheniformis* MW3 genome into the pKVM1 plasmid. After conjugation and homologous recombination, the *egfp* gene was inserted at the defined position on the genome of *B. licheniformis* MW3. To construct gene inversion mutant, the target gene were inversely cloned with their upstream and downstream regions and cloned into the pKVM1 plasmid. After conjugation and homologous recombination, the target gene was inversely inserted back to the same position on the genome of *B. licheniformis* MW3.

To construct the complementary strain for *yrkD*. the *yrkD* gene was assembled with the P43 promoter and ~1000 bp homologous sequence starting from 165 bp downstream of *pdo* gene into the pKVM1 plasmid. The P43-*yrkD* formed an expression cassette. The complementary strain for *nreBC* was constructed in a similar way, in which *nreB* and *nreC* were assembled into pKVM1 plasmid under the control of Pycec promoter with the same homologous sequence. After conjugation and single-crossover recombination, the whole plasmid with the indicated expression cassette was inserted to the indicated mutants at the site where is 165 bp downstream of *pdo* gene. The transcriptional direction of these expression cassettes was in the opposite of that of *pdo*.

**The sulfide or cysteine spiking test with *B. licheniformis* strains.** The resting cells of *B. licheniformis* strains were prepared according to a reported method with minor changes[10]. Briefly, cells of *B. licheniformis* strains were cultured to OD600 of ~0.8 in LB medium or the modified M9 medium at 37°C. Different inducers with defined concentrations were added twice into the cultures at 1-h intervals. Cells were harvested and washed once with HEPES buffer (50 mM, pH 7.0) containing 50 μM DTPA, 0.5 g/L NH4Cl, and 20 g/L glucose, and then suspended in the same buffer. 10 mL of the prepared resting cells at an OD600 of 2.0 was transferred to a 30-mL serum bottle with a rubber stopper. Freshly prepared NaHS or cysteine were added to initiate the reaction at the initial concentration of 400 μM or 600 μM, respectively. The samples were incubated at 37 °C with gently shaking. At various time intervals, the produced sulfide and total sulfane sulfur content were determined with methods described below.

**Detection of sulfide, cysteine, sulfite, thiosulfate, and total sulfane sulfur content.** The concentration of sulfide, cysteine, sulfite, and thiosulfate were

determined with the monobromobimane (mBBr) derivation method[10]. Briefly, 50 μL of culture supernatant was reacted with 5 μL of 25 mM mBBr at room temperature for 30 min in the dark, and then 110 μL of 10% acetic acid in acetonitrile was added to stop the reaction. The reaction mixture was centrifuged at 12,500 × g for 2 min. 70 μL of the supernatant was injected onto a C18 reverse phase column (VP-ODS, 150 × 4 mm, Shimadzu) and analyzed by using HPLC (LC-20AT, Shimadzu) with a fluorescence detector (RF-20A, Shimadzu). The column was maintained at 38 °C and eluted with a gradient of solution A (0.25% acetic acid) and solution B (100% methanol). A gradient program was used as follows: 0 to 1 min, 52.5% B; 1 to 15 min, 52.5 to 55% B; 15 to 15.1 min, 55 to 100% B; 15.1 to 20 min, 100% B; 20 to 20.1 min 100% to 7.5% B; 20.1 to 30 min, 7.5% B. The flow rate was 0.8 mL/min. The excitation wavelength and emission wavelength of the fluorescence detector were set at 370 nm and 485 nm.

The total sulfane sulfur content was reacted with sulfite to produce thiosulfate, which was detected as previously reported[65,66]. Cells were collected and washed with PBS buffer twice by centrifugation at 6000 × g for 5 min at 4 °C. Cell pellets from 1 mL of the suspension at OD600 of 3 were resuspended in 100 μL of the reaction buffer or control buffer and heated at 95 °C for 10 min. The reaction buffer contained 50 mM Tris-HCl buffer (pH 9.5) with 1% TritonX-100, 50 μM DTPA, and 1 mM sulfite, and the control buffer replaced sulfite with 0.5 mM DTT, which reduces sulfane sulfur to $H_2S$ to avoid the spontaneous oxidation of sulfane sulfur to thiosulfate. The samples were then centrifuged at 12,500 × g for 3 min at 4 °C, and 50 μL of the supernatant was incubated with 5 μL of 25 mM mBBr in acetonitrile at room temperature in the dark for 30 min. The reaction was stopped by adding 110 μL of 10% acetic acid in acetonitrile. After centrifugation at 12,500 × g for 10 min, 70 μL supernatant was analyzed by HPLC as mentioned above. The total sulfane sulfur was calculated by subtracting the thiosulfate in the control.

**RNA extraction and reverse-transcription quantitative PCR (RT-qPCR).** RNA samples from the cells of different strains were extracted by using the Trizol™ reagent according to the instructions (Thermo, US). As an optimization step, we noticed that the cell-broken efficiency was greatly enhanced by mixing the cells with 0.05% SDS (Sangon Biotech, China) and 20 mg/mL lysozyme (Solarbio, China) as the final concentrations for 20 min at 37 °C before treating with Trizol™. Total cDNA was synthesized using the All-In-One RT MasterMix (ABM). Residual DNA in the extracted RNA was further removed by using DNase I (Vazyme, China). Some of the gDNA-free RNA was saved for the detection of gDNA contamination.

RT-qPCR was performed with the BestarVR SybrGreen qPCR Mastermix (DBI) and the LightCycler 480II instrument (Roche) according to the specification. For calculation of relative expression levels of the target genes, *gyrB* was used as the internal standard, and the quantification method was the same as previously reported (Livak and Schmittgen, 2001). The thermocycler program for RT-qPCR was 94 °C for 5 min; 40 cycles at 94 °C for 5 s, 62 °C for 15 s, 72 °C 10 s; 72 °C for 7 min. The thermocycler program for melting curves was 94 °C for 2 min with the temperature being gradually decreased to 25 °C at 0.21 °C/s, and fluorescence acquisition was continuous.

**Promoter screening with an eGFP reporter plasmid in *E. coli*.** A promoter screening plasmid was constructed with *egfp* as a reporter gene. A 49 bp DNA sequence corresponding to terminator *TerB* and the *egfp* gene were constructed in the plasmid pBluescript SK- to get pSK::eGFP-Ter. The 10 intergenic regions were

cloned into the site between the terminator and *egfp* gene to get plasmids pSK::P1-eGFP-Ter to pSK::P10-eGFP-Ter, respectively. The constructs were introduced into *E. coli* strain MG1655. The *E. coli* cells harboring these ten plasmids were cultured until their $OD_{600}$ reached 0.4, then the cells were washed and resuspended in 50 mM Tris-HCl (pH 8.0) at an $OD_{600}$ of 1.0. The fluorescence intensity was recorded on a microplate spectrophotometer (BioTek, US), with the excitation wavelength at 485 nm and the emission wavelength at 528 nm. Three different colonies were cultured, and the fluorescence intensities were measured in the same way to get the average and standard derivation.

The intergenic regions that could activate the eGFP expression on the reporter plasmid were further truncated from its 5' end by 20-bp intervals. The truncated sequences were then introduced into the reporter plasmid to check the eGFP expression as above.

**5' rapid amplification of cDNA ends (5' RACE) analysis.** To determine the transcriptional start site for each identified promoter, 5' RACE was performed using the 5' RACE System for Rapid Amplification of cDNA Ends (version 2.0) as recommended by the manufacturer (Thermo Fisher, USA).

**The expression and purification of NreC and YrkD.** The *nreC* genes were amplified and cloned into pMAL-c2X to get pMalNreC. NreC were overexpressed and purified as MalE fusion proteins according to the manufacturer's instructions (NEB, UK). Fusion proteins were eluted in a Tris-HCl buffer (20 mM, pH 7.4) with 200 mM NaCl, 1 mM EDTA and 10 mM maltose.

The pET30 Ek/LIC plasmid was fused with *yrkD* to get pET30YrkD which was introduced into *E. coli* BL21(DE3) for overexpression. The recombinant *E. coli* strain was grown in LB medium at 37 °C with shaking until its $OD_{600}$ reached 0.6, and then 0.2 mM isopropyl-β-D-thiogalactopyranoside (IPTG) was added, and the cells were further cultivated at 25 °C, 150 rpm for 20 h. Cells were collected via centrifugation, washed twice with ice-cold lysis buffer (50 mM $NaH_2PO_4$, 200 mM NaCl, and 20 mM imidazole, pH 8.0), and broken through the high-pressure crusher SPCH-18 (STANSTED, UK). The recombinant YrkD with a His-tag was purified from *E. coli* BL21(DE3) via the Ni-NTA resin according to the specification as recommended by the manufacturer (Invitrogen, USA).

**The phosphorylation of NreC.** The NreC was phosphorylated according to a reported method[67]. Briefly, 20 μM purified NreC in 50 mM Tris–HCl, pH 8.0, was incubated with 200 mM acetyl phosphate (AcP), or as indicated in the text, at 25 °C for 30 min. After reaction, the residual AcP was removed by using a Zeba™ spin desalting column (Thermo Fisher, USA) according to the manual. The treated NreC was noted as phosphorylated NreC.

**Identifying the DNA binding sites of YrkD and NreC.** First, an electrophoretic mobility shift assay (EMSA) was used to detect the DNA binding site for YrkD and NreC. This method was modified from our previous study[27]. Briefly, the double-stranded DNA fragments were labeled with 5'-FAM (5-Carboxyfluorescein) at the 5' ends via PCR amplification and then used as the DNA probes for EMSA. A 15-μL EMSA reaction mixture containing 20 nM DNA probe, different amounts of phosphorylated NreC or purified YrkD, and 1× binding buffer (20 mM Tris-HCl, 20 mM KCl, 0.5 mM DTT, 2 mM EDTA, and 4% Ficoll-400, pH 8.0) were mixed and incubated at 25 °C for 30 min. The reaction mixture was then loaded onto a 6% native polyacrylamide gel and electrophoresed at 180 V for 2 h. The gel was photographed with a FluorChem Q system (Alpha Innotech).

The binding sites were identified in four steps. First, the intact intergenic regions (160 bp -474 bp) before *yrkD*, *nreC*, and *sqr* were used as the probes for EMSA to test their affinities with YrkD and NreC. Second, the intact intergenic regions were cut into several pieces (~110 bp per each) to locate the binding sites via EMSA. Third, the accurate binding sites were manually screened from the related DNA-binding sites of YrkD and NreC homologs in other bacteria[26,42]. Then, the DNA regions containing the predicted binding sites were cloned into pBluescript SK-, and the predicted binding sites were mutated by using a modified QuikChange method[68]. The related DNA regions were amplified and used as the probes for YrkD and NreC via EMSA analysis.

**Fluorescence polarization measurements.** Double-stranded 5' FAM-labeled DNA fragments containing the binding sites for YrkD and NreC were prepared in the same way as for the EMSA assay. 1 nM DNA was incubated with increasing amounts of different states of YrkD or NreC proteins (gradient dilution from an indicated concentration of protein) in a reaction buffer (50 mM Tris-HCl, pH 8.0) at 37 °C in the dark for 30 min as indicated in text. Fluorescence polarization measurements were conducted on a Synergy H1 Multi-Mode Reader (Biotek, US). All of the experiments were performed in triplicate, and the curves were fitted to deduce binding affinities by the GraphPad Prism 7.0 (GraphPad Software, US).

**Fluorescence assay to screen for inducers of YrkD and NreBC.** The *egfp* gene was inserted behind the *pdo* gene in *B. licheniformis* MW3 and the Δ*yrkD* mutant to construct A1109 and A1120 strains, respectively. The strains were grown in the modified M9 medium until the $OD_{600}$ reached about 0.8, and diverse inducers were

added. The varieties and concentrations of the inducers were indicated in the text. After incubating at 37 °C for defined time intervals, the cells were harvested by centrifugation and re-suspended in 50 mM HEPES buffer (pH 7.0) containing 50 μM DTPA, 0.5 g/L $NH_4Cl$, and 20 g/L glucose at an $OD_{600}$ of 1.0. Aliquots were transferred into a 96-well plate, and the eGFP fluorescence was measured by using the SynergyH1 microplate reader. The excitation wavelength was set at 485 nm, and the emission wavelength was set at 528 nm.

**Bioinformatics.** The genes encoding TCS were searched within 3 loci of the *sqr-pdo* operons in the 441 strains that we previously reported[10]. The 6 upstream and downstream genes from the operon were checked by using the conserved domain function from the NCBI website. The proteins marked with either two-component sensor histidine kinase or two-component response regulator were collected. Then, the phylogenetic trees by using DNA sequences of histidine kinases and response regulators were built. The 16 s rDNA sequences of the bacterial strains that harbor these TCSs were also collected and used for phylogenetic tree analysis. The PAS domains from these two-component sensor histidine kinases were phylogenetically analyzed with related PAS domains[69] and PAS domains that harbor heme cofactor[46]. Sequence alignment was done by using ClustalW, and the tree was built by using MEGA X with the Neighbor-Joining method and bootstrap at 1000 repeats, mainly according to the protocol suggested by Barry G. Hall[70].

**Statistics and reproducibility.** Experimental data are expressed as the mean ± standard deviation of the mean (SEM) of the number of tests stated for each experiment. All analysis was reproduced in at least three independent experiments. Statistical analysis was done using GraphPad Prism 9.0. The significant difference between the two groups was analyzed using an independent student's *t*-test; the *p*-value < 0.05 indicated statistical significance. Data in more than two groups were analyzed using independent one-way analysis of variance (ANOVA) to calculate the adjusted *p*-values; the *p*-value < 0.05 indicated statistical significance.

**Reporting summary.** Further information on research design is available in the Nature Portfolio Reporting Summary linked to this article.

## Data availability

The authors declare that the data supporting the findings of this study are available within the article (and its supplementary information files). Source data for all the figures are contained in Supplementary Data 4. Uncropped and unedited gel images are provided as Supplementary Fig. 14.

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

## Acknowledgements

We kindly thank Professor Chao Gao from Shandong University for providing the *B. licheniformis* strain MW3, plasmids and protocol used for gene manipulation. We also appreciate Professor Lushan Wang from Shandong University for providing the feathers collected from the poultry farm. We thank the support of Qilu Youth Scholar Startup Funding of SDU (to Y.X.) and the support from the State Key Laboratory of Microbial Technology. The work was supported by the Natural Science Foundation of China under grants number 31870085, 91951202, and 31961133015, respectively.

## Author contributions

Conceptualization, L.X. and Y.X.; methodology, Y.X. and Hu.L.; software, C.T. and Y.X.; validation, C.T., Y.S. and M.L.; investigation, C.T. and J.L.; resources, Y.X. and C.T.; data curation, C.T., and J.L.; writing—original draft preparation, J.L. and C.T.; writing—review and editing, L.X., Y.X., Ho.L. and Hu.L.; visualization, Y.X. and C.T.; supervision, L.X. and Y.X.; project administration, Y.X.; funding acquisition, Y.X. All authors have read and agreed to the published version of the manuscript.

## Competing interests

The authors declare no competing interests.
