## [Peer Review File · Communications Biology]

Reviewers' comments:

Reviewer #1 (Remarks to the Author):

Tang et al. report the regulation of a gene cluster involved in sulfur oxidation in *Bacillus licheniformis* MW3. They identified three promoters immediately upstream of the *yrkD*, *yrkE*, and *sqr* genes of the cluster members. The *yrkD* gene encodes a transcriptional factor, and they found that YrkD represses the *yrkD* and *yrkE* promoters and that the YrkD's repression is released by sulfane sulfur. The gene cluster includes the *nreBC* genes encoding a two-component regulatory system. The authors found that the NreBC regulatory system positively regulates the *sqr* promoter and that the *sqr* promoter is activated by sulfide such as NaHS. Sulfur oxidation is a physiologically important process for detoxifying cysteine and sulfide in all organisms. This process is required to be more accelerated in the bacteria industrially used for degrading cysteine-rich proteins such as keratin. Thus, elucidating the regulatory mechanism of the genes for sulfur oxidation in the relevant bacteria is entirely meaningful.

Major points

1) The authors suggested that the *sqr* promoter induction by sulfide is mediated through NreBC based on the data of Figs. 6 and 7. However, these data do not exclude the possibility that another regulator, in response to sulfide, activates or derepresses the *sqr* promoter. To confirm that the NreBC system is involved in the sulfide-dependent induction of the *sqr* promoter, a strain constructed by introducing the *nreBC* deletion into the M1109 strain is to be tested to determine whether the *sqr*-promoter induction by sulfide is abolished or declined. Alternatively, the *nreBC* expression level in the *sqr*-*pdo*-R and M1109 strains cultivated on sulfide is to be compared, and the relationship between the *nreBC* expression and the *sqr*-promoter activity in the presence and absence of sulfide should be determined.

2) The transcription start site from the *yrkE* promoter was determined using the transcript in the *E. coli* host but not *B. licheniformis* (Fig. 4E, lines 362–363). I do not think that all transcription start sites of *B. licheniformis* correspond entirely to those obtained in the *E. coli* host. The transcription start site from the *yrkE* promoter should be determined by the transcript of *B. licheniformis*.

Minor points

3) In many figures, including the supplemental materials, the size of letters and symbols presented are so small that they are difficult to decipher. They should all be improved for easy reading.

4) lines 119, 167, and 853: Since "eGFP" in these sentences refers to the gene but not the protein, they should be corrected as "*egfp*".

5) lines 240, 321, 334, 336, 372, 415, 418, 451, 876, 903: The gene names, such as *yrkD*, are to be italicized.

6) lines 138, 174, 193, 201, 213, 233, and 244: "pH =" -> "pH", "=" should be removed.

7) lines 20 and 43: "rich with" -> "rich in"

8) line 86: "mediums" -> "media"

9) line 88: "oligos" -> "oligonucleotides"

10) lines 98–102: " $\text{MgCl}_2 \cdot 6\text{H}_2\text{O}$ " -> " $\text{MgCl}_2 \cdot 6\text{H}_2\text{O}$ ", "0.01g/L" -> "0.01 g/L", "0.24g/L" -> "0.24 g/L", "chopped into" -> "chopped and added to". "Ampicillin" should not be italicized.

11) lines 139 and 245: "0.5 g of NH_4Cl , and 20 g of glucose per liter" -> "0.5 g/L

NH₄Cl, and 20 g/L glucose"

12) lines 156 and 212–214: The final concentrations of the respective compounds in the mixtures should be provided.

13) line 168: "encoding terminator" -> "corresponding to terminator"

14) line 171: "transformed into" -> "introduced into"

15) lines 189–204: What experiment was the purified NreB protein used for? The purities of NreB, NreC, and YrkD proteins should be confirmed by the methods such as SDS-PAGE, which should be presented in the supplemental materials. The sensor kinase of the TCS is often a membrane protein. Did you obtain the recombinant NreB-MBP protein in a soluble form?

16) line 275: I think that RHOD should be briefly explained.

17) lines 277–278: "a two-component regulatory system (TCS)" -> "TCS"

18) lines 284–285: "a two-component regulatory system (TCS), consisting of a sensor histidine kinase and a response regulator," -> "TCS"

19) line 309: "3 μ L 8 M" -> "3 μ L of 8 M"

20) lines 316, 332, and 844: "OD_{600nm}" -> "OD₆₀₀", "nm" is to be removed.

21) line 330: pH of the buffer should be provided.

22) lines 356–357: "the Neural Network Promoter Prediction webserver", a reference paper or a URL of the website should be provided.

23) line 366: The construction of the *sqr-pdo-R* strain should be mentioned here or in the materials and methods.

24) lines 390, 397, and 850: "RT-PCR", "qPCR"-> "RT-qPCR"

25) line 402: "(Fig. 7F)" -> "(Fig. 6F)"?

26) line 408: "introduce MW3" -> "induce MW3"?

27) line 409: "yrkD yrkH" -> "yrkD, yrkH", add comma.

28) line 438: "Kd", "d" should be subscripted.

29) line 440: The method of phosphorylation of NreC should be explained, along with a reference citation here or in the materials and methods.

30) lines 447–449: "its encoding gene yrkD results in" -> "its gene yrkD to give"? "lead to a transcript" -> "leading to a transcript"

31) lines 475–479 and 494: I think that it is not appropriate to refer to the PAS domain of NreB as NreB^{PAS}.

32) lines 481–496: Is it experimentally confirmed that NreB contains heme iron?

33) lines 484–485: "Fe²⁺", "O²⁻", "2⁺" and "2⁻" are to be superscripted.

34) lines 869–871: I cannot find symbols "+" and "-" in Fig. 5B.

35) Fig. 9F: I think that "-" and "+" of YrkD in lanes IR2-2* and control are incorrect.

36) Fig. 9G and H: The mutated bases in the mIR1-6, mIR2-2, and mIR9 should be indicated.

37) Supplementary Table S2: The boundaries of each oligonucleotide sequence are indistinguishable, which should be improved.

Reviewer #2 (Remarks to the Author):

In this manuscript, authors elaborately verified the structure of a gene cluster associated with sulfur metabolism, including its encoded one repressor YrkD, one enhancer NreBC and three promoters. The whole regulation process of gene expression of this gene cluster was well described with solid details, from sensing sulfane sulfur and sulfide, YrkD and NreBC DNA binding sites and finally different genes expression levels. All results are credible. While similar gene clusters and the functions of genes in this cluster from other bacteria had been reported before. What most attractive part of this manuscript is that *B. licheniformis* is a feather degrading bacterium FDB. Its sulfur metabolism pathway is greatly related to its feather degradation. It is sorry to see the description of relationship between this gene cluster and feather degradation is very weak in this manuscript. Actually, this gene cluster is not only associated with sulfur detoxification, but also the extracellular keratin disulfide bond reduction. It's very critical for FDBs to produce reducing agents for extracellular disulfide reduction to help its feather degradation and survive on feather, though they are toxic to bacterium itself and need to be detoxification. The logic between the function of this gene cluster and feather degradation is blur. The author should clearly declare the feather degradation process in Fig. 10 to tell readers what function of this gene cluster's expression regulation in this bacterium's feather utilization, including how feather is degraded to yield cystine, how cystine is transformed to H₂S and sulfane sulfur. H₂S and sulfite's exportation is important for these reducing agents' detoxification and helpful for keratin's disulfide bond reduction. Describing the function and regulation mechanism of this gene cluster only reveal small aspect of bacterium's sulfur metabolism pathway. After reading this manuscript, I do not know how the mechanism revealed in this paper can help the further genetic modification of this bacterium to improve its feather degradation efficiency. Some typo errors need to be revised, such as in line 402, it should be Fig. 6F, not Fig. 7F.

Reviewer #3 (Remarks to the Author):

In the present work, Tang et al identified and characterized a sulfur-responding gene cluster in *B. licheniformis*, which contains three promoters with the first two repressed by YrkD and last one activated by NreBC. YrkD sensed sulfane sulfur and derepressed all genes in the cluster. NreBC responded to sulfide and amplified the transcription of *sqr* and *pdo*. The two systems co-controlled the expression of the gene cluster, which helps the bacterium to grow on feather. Overall, many experiments were well designed and the findings were novel, but lack some important complementations and controls. Most EMSA data are not convincing.

1. The complementary strains were missing for all mutants. Without this, it's hard to verify the link between specific genes and their phenotypes. This shouldn't be hard, since the authors have identified their promoters.
2. In Figure 9C, no clear binding between NreC and IR9. Same issue for Figure 9E-H. Cold competition with dI-dC should be added. Negative controls were missing as well for all bindings.
3. Statistical analyses were missing in all line charts, such as Figure 2B-F, 3A-D, 5B, and 6CE.
4. Gene names should be italic in all figures or texts. Some typo such as 'strian'.

Responses to comments from Referees: We sincerely thank the editor and the reviewers for your valuable feedback. The suggestions have been incorporated into the revised manuscript. A point-by-point response is given as follows:

Reviewers' comments:

Reviewer #1 (Remarks to the Author):

Major points

1) The authors suggested that the *sqr* promoter induction by sulfide is mediated through NreBC based on the data of Figs. 6 and 7. However, these data do not exclude the possibility that another regulator, in response to sulfide, activates or derepresses the *sqr* promoter. To confirm that the NreBC system is involved in the sulfide-dependent induction of the *sqr* promoter, a strain constructed by introducing the *nreBC* deletion into the M1109 strain is to be tested to determine whether the *sqr*-promoter induction by sulfide is abolished or declined. Alternatively, the *nreBC* expression level in the *sqr*-pdo-R and M1109 strains cultivated on sulfide is to be compared, and the relationship between the *nreBC* expression and the *sqr*-promoter activity in the presence and absence of sulfide should be determined.

Response: The function of NreBC to upregulate *sqr* is supported with the wild type, its *nreBC*-deletion mutant, and the complementation strain

***ΔnreBC::nreBC*. To further support the role of NreBC, the *nreBC* genes were deleted from M1109 to generate M1109-*ΔnreBC*. The transcription of *sqr* was abolished in M1109-*ΔnreBC* even when sulfide was used as an inducer (Fig. S7).**

The result was added to the manuscript (new lines 453 -457).

2) The transcription start site from the *yrkE* promoter was determined using the transcript in the *E. coli* host but not *B. licheniformis* (Fig. 4E, lines 362–363). I do not think that all transcription start sites of *B. licheniformis* correspond entirely to those obtained in the *E. coli* host. The transcription start site from the *yrkE* promoter should be determined by the transcript of *B. licheniformis*.

Response: We have tried to find the transcription start site of the *yrkE* promoter

in *B. licheniformis* MW3, but no confident results were obtained. Since no terminator sequence was found between promoters before *yrkD* and *yrkE*, it is most probably that the transcription from the promoter was weak. Its role is likely for the leaky expression of the genes in the cluster besides *yrkD*. In the presence of elemental sulfur, the expression is mainly from the first promoter. This explanation was given in the main text (new lines 403-408).

Minor points

3) In many figures, including the supplemental materials, the size of letters and symbols presented are so small that they are difficult to decipher. They should all be improved for easy reading.

Response: We enlarged the letters and symbols in all figures except figure 1, figure 10, and figure S14.

4) lines 119, 167, and 853: Since "eGFP" in these sentences refers to the gene but not the protein, they should be corrected as "*egfp*".

Response: Thank you. "eGFP" was changed to the "*egfp*" in the text when it refers to the gene.

5) lines 240, 321, 334, 336, 372, 415, 418, 451, 876, 903: The gene names, such as *yrkD*, are to be italicized.

Response: Thank you. The gene names were italicized in the referred lines.

6) lines 138, 174, 193, 201, 213, 233, and 244: "pH =" -> "pH", "=" should be removed.

Response: "pH =" was changed to the "pH" in new lines 153, 189, 208, 215, 235, 254, 266.

7) lines 20 and 43: "rich with" -> "rich in"

Response: The sentences were changed to "Feather is rich in cysteine" in new

line 19 and “Feather keratin is rich in cysteine residues” in new line 39.

8) line 86: "mediums" -> "media"

Response: It was changed to the sentence “Bacterial strains, plasmids, culture media, and chemicals” in new line 87.

9) line 88: "oligos" -> "oligonucleotides"

Response: The sentence was changed to “The used oligonucleotides and their usages are listed in Table S2” in new line 89.

10) lines 98–102: "MgCl₂6H₂O" -> "MgCl₂•6H₂O", "0.01g/L" -> "0.01 g/L", "0.24g/L" -> "0.24 g/L", "chopped into" -> "chopped and added to". "Ampicillin" should not be italicized.

Response: The sentence was changed to “and it contained of 0.5 g/L NH₄Cl, 0.5 g/L NaCl, 0.3 g/L K₂HPO₄, 0.4 g/L KH₂PO₄, 0.24 g/L MgCl₂•6H₂O, 0.1 g/L yeast extract, 0.01 g/L CaCl₂, 0.24 g/L MgSO₄, 1% (w/v) chopped feather, pH 7.5” in new lines 98-100.

The sentence was changed to “The feathers were chopped and added to the medium. Ampicillin (Amp) and Kanamycin (Kan) were used at 100 µg/mL and 50 µg/mL” in new lines 102-103.

11) lines 139 and 245: "0.5 g of NH₄Cl, and 20 g of glucose per liter" -> "0.5 g/L NH₄Cl, and 20 g/L glucose"

Response: It was changed to “0.5 g/L NH₄Cl, and 20 g/L glucose” in new lines 154 and 267.

12) lines 156 and 212–214: The final concentrations of the respective compounds in the mixtures should be provided.

Response: We provided the final concentrations of the respective compounds in new lines 171 and 232-235.

13) line 168: "encoding terminator" -> "corresponding to terminator"

Response: The sentence was changed to “A 49 bp DNA sequence corresponding to terminator TerB” in new line 183.

14) line 171: "transformed into" -> "introduced into"

Response: The sentence was changed to “The constructs were introduced into *E. coli* strain MG1655” in new line 187.

15) lines 189–204: What experiment was the purified NreB protein used for? The purities of NreB, NreC, and YrkD proteins should be confirmed by the methods such as SDS-PAGE, which should be presented in the supplemental materials. The sensor kinase of the TCS is often a membrane protein. Did you obtain the recombinant NreB-MBP protein in a soluble form?

Response: We did not report the purified NreB in this manuscript. Indeed, sensor kinase in most two-component systems is membrane protein, but it is not the case for NreB. As we mentioned in the discussion, the NreB is homologous to NreB from *Staphylococcus carnosus* except its PAS domain. The NreB from *S. carnosus* has been proved to be in soluble form (Kamps, et al., *Molecular Microbiology*, 2004). The recombinant NreB-MBP is soluble.

16) line 275: I think that RHOD should be briefly explained.

Response: A brief explanation of RHOD was given (new lines 56, 303-304).

17) lines 277–278: "a two-component regulatory system (TCS)" -> "TCS"

Response: It was changed to “TCS” in new line 306.

18) lines 284–285: "a two-component regulatory system (TCS), consisting of a sensor histidine kinase and a response regulator," -> "TCS"

Response: It was changed to “a TCS, consisting of a sensor histidine kinase and a

response regulator” in new lines 306-307.

19) line 309: "3 μ L 8 M" -> "3 μ L of 8 M"

Response: The sentence was changed to “On LB plates with 3 μ L of 8 M cysteine plotted in middle” in new line 340.

20) lines 316, 332, and 844: "OD_{600nm}" -> "OD₆₀₀", "nm" is to be removed.

Response: It was changed to “OD₆₀₀” in new lines 346, 365 and 914.

21) line 330: pH of the buffer should be provided.

Response: The pH of the buffer was provided in new line 363.

22) lines 356–357: "the Neural Network Promoter Prediction webserver", a reference paper or a URL of the website should be provided.

Response: A URL of the website was provided in new line 398.

23) line 366: The construction of the *sqr-pdo*-R strain should be mentioned here or in the materials and methods.

Response: The detailed procedure to construct the *sqr-pdo*-R strain was added into the materials and methods section (new lines 133-136).

24) lines 390, 397, and 850: "RT-PCR", "qPCR"-> "RT-qPCR"

Response: It was changed to “RT-qPCR” in new lines 436, 445 and 924.

25) line 402: "(Fig. 7F)" -> "(Fig. 6F)"?

Response: Yes, it should be the Figure 6F in new line 437.

26) line 408: "introduce MW3" -> "induce MW3"?

Response: Yes, the sentence was changed to “When 10 μ M to 500 μ M elemental sulfur was used to induce MW3 for 10 min” in new line 459.

27) line 409: "*yrkD yrkH*" -> "*yrkD, yrkH*", add comma.

Response: The comma in "*yrkD, yrkH*" was added in new line 460.

28) line 438: "Kd", "d" should be subscripted.

Response: The "d" was subscripted in new line 492.

29) line 440: The method of phosphorylation of NreC should be explained, along with a reference citation here or in the materials and methods.

Response: The detailed method of phosphorylation of NreC was added in materials and methods, and the reference was cited (new lines 220-225).

30) lines 447–449: "its encoding gene *yrkD* results in" -> "its gene *yrkD* to give"?
"lead to a transcript" -> "leading to a transcript"

Response: The sentence was changed to “one starts before its gene *yrkD* to give a transcript of all the 11 genes, and the other starts after *yrkD* leading to a transcript of the rest 10 genes” in new lines 501-503.

31) lines 475–479 and 494: I think that it is not appropriate to refer to the PAS domain of NreB as NreB^{PAS}.

Response: All ‘NreB^{PAS}’ used in the main text and the supplementary text were changed into ‘the PAS domain of NreB’.

32) lines 481–496: Is it experimentally confirmed that NreB contains heme iron?

Response: In this paper, we gave a prediction. The experiment proof was not reported in this manuscript. However, we have proved NreB contains heme iron by LC-MS/MS method, and plan to report these results in the next paper which concerns more about the H₂S sensing mechanism.

33) lines 484–485: "Fe²⁺", "O²⁻", "2⁺" and "2⁻" are to be superscripted.

Response: The "2+" and "2-" were superscripted in lines 538–539.

34) lines 869–871: I cannot find symbols "+" and "-" in Fig. 5B.

Response: The symbol "+" and "-" was removed from this figure legend. If no inducer was used, only strain's name was shown in graph legends.

35) Fig. 9F: I think that "-" and "+" of YrkD in lanes IR2-2* and control are incorrect.

Response: The experiments were repeated, the figures were reorganized and the symbols "-" and "+" used for lanes IR2-2* and control were corrected.

36) Fig. 9G and H: The mutated bases in the mIR1-6, mIR2-2, and mIR9 should be indicated.

Response: The mutated bases of the mIR1-6, mIR2-2, and mIR9 were indicated in new Supplementary Figure S12.

37) Supplementary Table S2: The boundaries of each oligonucleotide sequence are indistinguishable, which should be improved.

Response: The boundaries of each oligonucleotide sequence with solid lines were added in new Supplementary Table S2.

Reviewer #2 (Remarks to the Author):

In this manuscript, authors elaborately verified the structure of a gene cluster associated with sulfur metabolism, including its encoded one repressor YrkD, one enhancer NreBC and three promoters. The whole regulation process of gene expression of this gene cluster was well described with solid details, from sensing sulfane sulfur and sulfide, YrkD and NreBC DNA binding sites and finally different genes expression levels. All results are credible. While similar gene clusters and the functions of genes in this cluster from other bacteria had been reported before. What

most attractive part of this manuscript is that *B. licheniformis* is a feather degrading bacterium FDB. Its sulfur metabolism pathway is greatly related to its feather degradation. It is sorry to see the description of relationship between this gene cluster and feather degradation is very weak in this manuscript. Actually, this gene cluster is not only associated with sulfur detoxification, but also the extracellular keratin disulfide bond reduction. It's very critical for FDBs to produce reducing agents for extracellular disulfide reduction to help its feather degradation and survive on feather, though they are toxic to bacterium itself and need to be detoxification. The logic between the function of this gene cluster and feather degradation is blur. The author should clearly declare the feather degradation process in Fig. 10 to tell readers what function of this gene cluster's expression regulation in this bacterium's feather utilization, including how feather is degraded to yield cystine, how cystine is transformed to H₂S and sulfane sulfur. H₂S and sulfite's exportation is important for these reducing agents' detoxification and helpful for keratin's disulfide bond reduction. Describing the function and regulation mechanism of this gene cluster only reveal small aspect of bacterium's sulfur metabolism pathway. After reading this manuscript, I do not know how the mechanism revealed in this paper can help the further genetic modification of this bacterium to improve its feather degradation efficiency.

Response: After reading your comments, we realized that we should clearly mention the novelty of this report in the introduction. We added the information in the introduction section (new lines 79-84). First, we report a sulfur-oxidizing operon that is regulated by two regulators. Previously reported operons are all regulated by a single regulator. Second, the two regulators synergistically regulate the operon. One of the regulators is the same as previously reported to sense sulfane sulfur, and the other senses sulfide. A sulfide-sensor has not previously reported to regulator sulfur oxidizing genes. Third, the gene cluster is required for the growth of the bacteria on feathers, as it is shown to involve in detoxifying sulfide and sulfur sulfur derived from feather degradation.

As for how the bacterium degrades feather, it is beyond the scope of this

report. It should be a separate project. However, we did observe the accumulation of cysteine in the medium during feather degradation. A recent report (ref. 6 of our manuscript) has also observed cysteine accumulation and suggested that cysteine may be involved in breaking the disulfide bonds in feathers. The authors suggested that further research is needed to test the hypothesis. We agree and believe that further investigation on the topic is required. The information and discussion are added to the manuscript (new lines 41-42; 587-592).

Some typo errors need to be revised, such as in line 402, it should be Fig. 6F, not Fig. 7F.

Response: The typo errors were carefully checked and corrected in new line 437.

Reviewer #3 (Remarks to the Author):

In the present work, Tang et al identified and characterized a sulfur-responding gene cluster in *B. licheniformis*, which contains three promoters with the first two repressed by YrkD and last one activated by NreBC. YrkD sensed sulfane sulfur and derepressed all genes in the cluster. NreBC responded to sulfide and amplified the transcription of *sqr* and *pdo*. The two systems co-controlled the expression of the gene cluster, which helps the bacterium to grow on feather. Overall, many experiments were well designed and the findings were novel, but lack some important complementations and controls. Most EMSA data are not convincing.

1. The complementary strains were missing for all mutants. Without this, it's hard to verify the link between specific genes and their phenotypes. This shouldn't be hard, since the authors have identified their promoters.

Response: In this paper, we mainly discussed that YrkD and NreBC can

synergistically regulate the expression of sulfide oxidation genes, which helps the bacterium to grow on feather. Therefore, we complemented *yrkD* and *nreBC* by inserting these two genes with indicated constitutive promoters at the downstream of the *pdo* gene. Compared with $\Delta yrkD$, the ability of $\Delta yrkD::yrkD$ to oxidize sulfide can be restored to the wild type MW3 (Fig. 3E). After *nreBC* was complemented, the $\Delta nreBC::nreBC$ strain could be induced by elemental sulfur as wild type (Fig. 3F). These results were added in main text (new lines 371-378).

2. In Figure 9C, no clear binding between NreC and IR9. Same issue for Figure 9E-H. Cold competition with dI-dC should be added. Negative controls were missing as well for all bindings.

Response: Thanks for your reminder. The no clear binding is due to low affinity of non-phosphorylated NreC to IR9. As we found that phosphorylated NreC could enhance its binding ability to IR9 (new Fig. S10, Fig. 9C&D). For new Fig. 9D&G, ~6.6 ng/ \square L poly(dI-dC) was added, and a fragment within *yrkD* (131 bp) and a fragment within *NreC* (147 bp) were used as the negative control for nonspecific binding. These results could prove that the bindings of NreC to IR9 and YrkD to IR1-6 and IR2-2 were specific. In addition, mIR1-6, mIR2-2 and mIR9 with mutated binding sites also could not bind to YrkD and NreC, respectively (Fig. 9I&J). The fragments of mIR1-6, mIR2-2 and mIR9 could also be considered as negative controls. Besides, we also tried to determine their binding affinities by using fluorescence polarization test. The result also showed that their K_d have physiological significance.

3. Statistical analyses were missing in all line charts, such as Figure 2B-F, 3A-D, 5B, and 6CE.

Response: The statistical analyses were carefully performed and added to figures except for growth curves. The statistical methods were added in indicated figure

legends. We think the differences among wild type and mutant strains are clear in these growth curves, and many articles published in journals under the Nature press also did not perform statistical analyses for growth curves (González-Magaña, et al, Communications Biology, 2022 Nov 5;5(1):1189; Hsueh, et al, Nat Microbiol, 2022, 7(8):1210-1220; Sankari, et al, Nat Microbiol, 2022 Sep;7(9):1453-1465.).

4. Gene names should be italic in all figures or texts. Some typo such as 'strian'.

Response: Gene names were italicized in all figures and texts. The other typo errors were carefully checked and corrected.

REVIEWERS' COMMENTS:

Reviewer #1 (Remarks to the Author):

Most concerns that I had have been addressed by revisions and an additional experiment. I found additional minor points to be modified as follows.

- 1) line 59: "a new gaseous signaling molecule" -> "a gaseous signaling molecule"
- 2) line 82: "whereas, previously reported regulators" -> "whereas previously reported regulators"
Please delete the comma.
- 3) line 103: "Kanamycin" -> "kanamycin" "k" should be in lowercase.
- 4) line 110: "Sangon Biotech. (China)" -> "Sangon Biotech (China)" Please delete the period.
- 5) line 167: "real-time quantitative PCR" -> "real-time quantitative PCR (RT-qPCR)"
- 6) line 210: "transformed into" -> "introduced into"
- 7) line 222: "200 mM acetyl phosphate (AcP)" Is this a final concentration? Did the reaction mixture contain any buffer, such as 50 mM Tris-HCl (pH 8.0)? They should be clarified.
- 8) lines 233–235: ", and 5× binding buffer (100 mM Tris-HCl, 100 mM KCl, 2.5 mM DTT, 10 mM EDTA, and 20% Ficoll-400, pH 8.0) were mixed and incubated at 25 °C for 30 min." You mean "in a binding buffer (20 mM Tris-HCl, 20 mM KCl, 0.5 mM DTT, 2 mM EDTA, and 4% Ficoll-400, pH 8.0) was incubated at 25 °C for 30 min."?
- 9) line 279–280: "two-component systems" -> "TCSs"
- 10) line 301: "yrkG" is not found in Figs 1 and 10. There is no "yrkJ" in the text.
- 11) line 312: "TCS, were" -> "TCS were" Please delete the comma.
- 12) line 347: "from at 5 min" -> "at 5 min"
- 13) line 349: "Bothe" -> "Both"
- 14) line 385: "the induced B. licheniformis MW3" What substance was used to induce the transcription of MW3? Elemental sulfur or sulfide? This should be mentioned.
- 15) line 405: "yrkE" should be italicized.
- 16) lines 407–408: "The transcription from the promoter was properly for the leaky expression of the genes in the cluster besides yrkD." You mean, "The transcription from the yrkD promoter might cause the leaky expression of the genes in the cluster besides yrkD."?
- 17) line 409: "To confirm the orientation of promoter in IR9" You mean "To confirm that the promoter in IR9 independently works"?
- 18) line 412: "the TIS in IR9 is specifical for" -> "the promoter in IR9 is specific for"?
- 19) lines 506–507: "also transcribed within the two transcripts" -> "also transcribed from the more distal promoters"?
- 20) lines 513–514: "the reversed sqr and pdo in the sqr-pdo-R strain were sufficiently induced by sulfide (Fig. 5B)." You mean "the induction profile of sqr and

pdo in the *sqr-pdo*-R strain was almost the same as that of MW3 (Figs. 3A&5B).”?

21) line 594: “Cys” -> “cysteine”

22) lines 601–602: The result of the Δ *yrkD* mutant in Fig. 2E should also be mentioned here. Moreover, you should discuss the reason why the growth of the Δ *nreBC* mutant was not significantly affected by 40 μ M sulfane sulfur (Fig. 2F).

23) line 918: “RT-qPCR” The data of Fig. 4A appears to be obtained by “reverse transcription PCR” but not “real-time qPCR.”

24) line 921: “eGFP” -> “*egfp*” (italicized)

25) line 936–937: “*B. licheniformis* MW3 and *sqr-pdo*-R were” -> “The *sqr-pdo*-R strain was”

26) Figs. 6F, 7B–E, 8F, and S7B: “Flod of control” -> “Fold of control” (Their vertical axes)

27) line 954: “(white)” -> “(light)”

28) line 996: “NreC-Pi” -> “YrkD”?

Reviewer #3 (Remarks to the Author):

The authors addressed the previous comments from the reviewer.